# Crystal orientation fabric anisotropy causes directional hardening of the Northeast Greenland Ice Stream

Tamara Annina Gerber [1] ✉, David A. Lilien [2], Nicholas Mossor Rathmann[1], Steven Franke [3], Tun Jan Young [4,5], Fernando Valero-Delgado[3], M. Reza Ershadi[6], Reinhard Drews [6], Ole Zeising [3], Angelika Humbert [3,7], Nicolas Stoll [3,7], Ilka Weikusat [3,6], Aslak Grinsted [1], Christine Schøtt Hvidberg [1], Daniela Jansen [3], Heinrich Miller[3], Veit Helm [3], Daniel Steinhage [3], Charles O'Neill [8], John Paden [9], Siva Prasad Gogineni[10], Dorthe Dahl-Jensen[1,2] & Olaf Eisen [3,7] ✉

The dynamic mass loss of ice sheets constitutes one of the biggest uncertainties in projections of ice-sheet evolution. One central, understudied aspect of ice flow is how the bulk orientation of the crystal orientation fabric translates to the mechanical anisotropy of ice. Here we show the spatial distribution of the depth-averaged horizontal anisotropy and corresponding directional flow-enhancement factors covering a large area of the Northeast Greenland Ice Stream onset. Our results are based on airborne and ground-based radar surveys, ice-core observations, and numerical ice-flow modelling. They show a strong spatial variability of the horizontal anisotropy and a rapid crystal reorganisation on the order of hundreds of years coinciding with the ice-stream geometry. Compared to isotropic ice, parts of the ice stream are found to be more than one order of magnitude harder for along-flow extension/ compression while the shear margins are potentially softened by a factor of two for horizontal-shear deformation.

The orientation of the crystals that comprise glacier ice exerts an important physical control over its bulk mechanical properties[1,2]. In the absence of strain, the crystal orientation fabric (COF) of snow and ice tends to be close to isotropic, so the c-axes point in random directions. In many parts of ice sheets, however, deformation causes an apparent rotation of c-axes towards the maximum shortening direction[3,4], resulting in an anisotropic COF that can be overprinted by subsequent deformation events[5]. In other words, the COF reflects past ice deformation while simultaneously affecting the present-day mechanical properties.

Most state-of-the-art large-scale ice-flow models either ignore the mechanical anisotropy of ice entirely[6] or infer isotropic enhancement factors that subsume some effect of anisotropy[7,8]. These enhancement factors are often tuned as model parameters or based on experimental values. In both cases, a physical understanding of the spatial and temporal variation of COF anisotropy is missing[8]. Anisotropy may explain some of the major discrepancies between modelled and observed surface velocities in highly dynamic areas such as ice streams[9,10] since the commonly used isotropic flow law is no longer valid when the COF is anisotropic. The resulting mechanical anisotropy

[1]Section for the Physics of Ice, Climate and Earth, Niels Bohr Institute, University of Copenhagen, Copenhagen, Denmark. [2]Centre for Earth Observation Science, University of Manitoba, Winnipeg, MB, Canada. [3]Alfred Wegener Institute, Helmholtz Centre for Polar and Marine Research, Bremerhaven, Germany. [4]Scott Polar Research Institute, University of Cambridge, Cambridge, United Kingdom. [5]School of Geography & Sustainable Development, University of St Andrews, St Andrews KY16 9AL, United Kingdom. [6]Department of Geosciences, Tübingen University, Tübingen, Germany. [7]Department of Geosciences, University of Bremen, Bremen, Germany. [8]EH Group Inc., Tuscaloosa, AL, USA. [9]Centre for Remote Sensing and Integrated Systems (CReSIS), University of Kansas, Lawrence, KS, USA. [10]Remote Sensing Centre, University of Alabama, Tuscaloosa, AL, USA. ✉e-mail: tamara.gerber@nbi.ku.dk; olaf.eisen@awi.de

potentially introduces errors of unknown magnitude in modelled strain rates and, thus, bias in basal sliding velocities obtained by inversion methods[7].

Ice streams are the primary source of dynamic ice mass loss and are hence of great importance in the stability of ice sheets[11–13]. Accordingly, it is crucial to accurately reproduce ice stream dynamics in prognostic ice-sheet models to reduce uncertainties in estimates of future ice-sheet evolution and sea-level rise. In regions of fast flow (> 10 ma$^{-1}$) with lateral compression and along-flow extension through flow acceleration, the development of strong, spatially variable COFs is expected and might facilitate streaming ice[14]. A number of ice-flow models capable of simulating COF evolution or its effects on ice deformation[15] as well as combining these two mechanisms[16,17] were already developed more than a decade ago. However, both the application of these models to ice streams and spatially extensive in-situ observations of the COF in these dynamic regions remain rare.

Direct observations of COFs in ice cores are limited to point measurements and by the unknown orientation of the core. Furthermore, most deep ice cores are drilled at ice domes or at ice divides, where the ice tends to flow slowly, and the COF is primarily a result of uniaxial or confined vertical compression[18]. These sites are unrepresentative of the COF in more dynamic areas, such as ice streams and their immediate surrounding where the strain history is more complex.

Radio-echo-sounding (RES) surveys are widely used in glaciology for mapping ice thicknesses, basal properties, and internal stratigraphy with electromagnetic waves[19–22]. Single ice crystals show dielectric anisotropy due to their uniaxial birefringent property, i.e. the relative dielectric permittivity is larger in the direction of their c-axis than in the direction of their basal plane. The COF of polycrystalline ice determines its bulk relative dielectric permittivity and affects the speed of electromagnetic wave propagation. Horizontal anisotropy modulates RES signals[23,24] such that the propagation speed of electromagnetic waves depends on their polarisation relative to the preferred orientation of the c-axes and techniques have recently been developed to derive information on the horizontal component of the COF from RES[25–28]. In areas where a horizontally anisotropic COF prevails, two different effects are observed in radargrams that can be related to the strength of horizontal anisotropy: (1) Interference of wave components travelling at different speeds leads to nodes of maximum/minimum radar return power[23] termed here as birefringence-induced beat signature (Fig. 1c, d) and (2) travel-time differences, $\Delta t$, for waves reflected off the same internal layers under orthogonal antenna polarisations due to differences in propagation speed (Fig. 1e, f)[28]. The techniques used to derive the horizontal anisotropy from these two effects are detailed in the Supplementary Information Sections 1.1 and 1.3.

In this study, we use the characteristic radar signals induced by the COF to determine the distribution of the horizontal anisotropy at the onset of the Northeast Greenland ice stream (NEGIS), the largest ice stream in Greenland. Using an extensive data set of radar measurements with various radar systems and a combination of independent methods, we determine the distribution of anisotropy in the horizontal plane. Previous studies[25,27,28] have used similar methods in isolation to infer the horizontal COF anisotropy of ice streams from radar data. The application of such methods on a large spatial scale and the comparison between different radar methods, COF-evolution modelling, and ice-core data allows us to obtain a robust understanding of the COF distribution and to constrain mechanical anisotropy by calculating directional flow-enhancement factors. Our results show a spatially variable horizontal COF anisotropy that considerably affects the ice stiffness in the NEGIS onset depending on the deformation mechanism and its direction relative to the COF principal axes.

## Results

We analysed the birefringence-induced beat signature and travel-time differences of radar waves using a composite of different radio-glaciological data sets. Our survey region covers an area of approximately 24,000 km$^2$ and encompasses the fast-flowing central part as

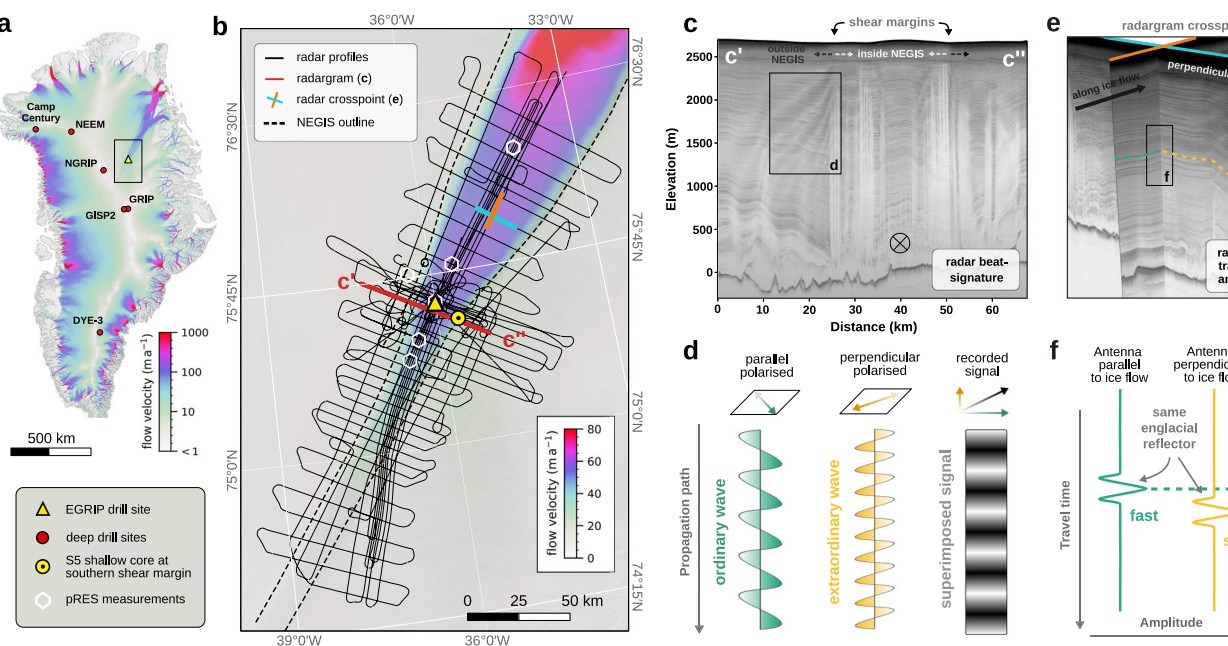

**Fig. 1 | Overview of the study area and radar methods. a** Surface ice-flow velocities of the Greenland ice sheet[64] with the locations of deep ice-core drill sites and the outline of the study area. **b** Onset region of the Northeast Greenland ice stream (NEGIS) and collected data. **c** Example of an airborne radargram crossing the ice stream near the EGRIP (East Greenland ice-core project) ice coring site, showing internal birefringence-induced extinction node lines particularly pronounced near the shear margins. As illustrated in panel **d**, the birefringence power extinction nodes arise in horizontally anisotropic ice through the interference of two orthogonal radar wave components travelling at slightly different wave speeds. **e** Two intersecting radar profiles, and **f** schematic example of travel-time differences of internal reflections.

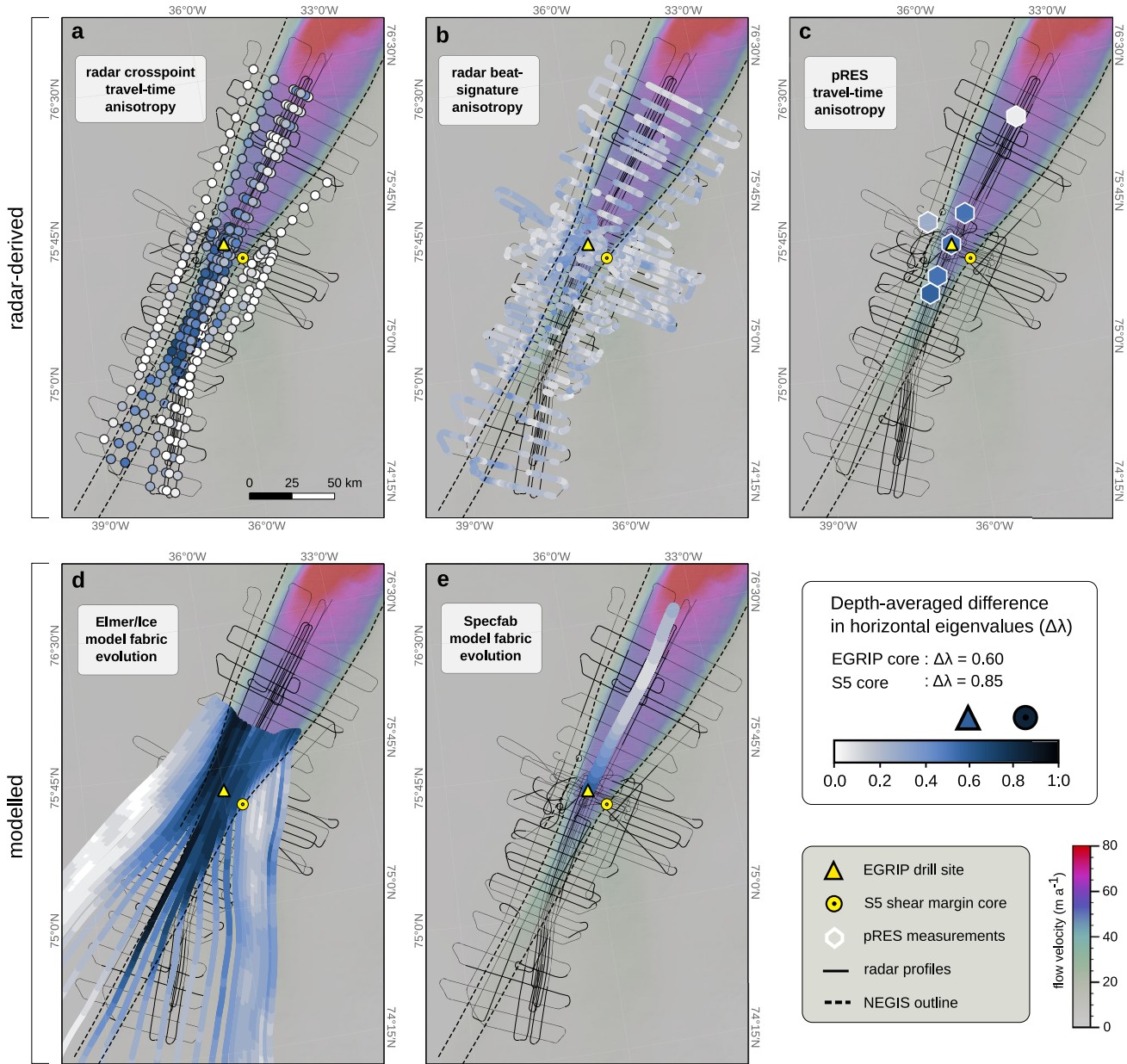

**Fig. 2 | Spatial distribution of horizontal crystal orientation fabric (COF) anisotropy.** Depth-averaged difference in horizontal eigenvalues ($\Delta\lambda$) inferred by **a** radar crosspoint travel-time analysis, **b** radar beat-signature analysis, **c** phase-sensitive radio-echo-sounding (pRES) travel-time analysis, and modelled by a **d** COF evolution model implemented in Elmer/Ice, and **e** Specfab COF evolution model along a flow line. Both the travel-time analysis of airborne and pRES radar as well as the beat-signature approach, are sensitive to the COF orientation relative to the antenna orientation and should thus be regarded as lower-bound limits, while the results obtained from COF evolution models can be regarded as absolute values. The background shows the satellite-based surface flow velocities[64].

well as the shear margins of the NEGIS onset (Fig. 1a, b). The data comprise an extensive airborne ultra-wideband radar survey[29], a ground-based radar profile across the northwestern shear margin near the East Greenland ice-core project (EGRIP) deep ice-core site, as well as polarimetric phase-sensitive radio-echo-sounding (pRES) measurements at six distinct locations (Fig. 1b). We derived the horizontal anisotropy from travel-time differences in polarimetric pRES measurements (Fig. 2c) and at airborne radar crosspoints with approximately orthogonal antenna polarisations (Fig. 2a), as well as by analysing the frequency of the beat signature in airborne and ground-based radar profiles (Fig. 2b). By combining these radar data sets and complementing methods, we obtain a spatially extensive, high-resolution and robust understanding of the distribution of the depth-averaged anisotropy in the horizontal plane in an active ice

stream. For a comprehensive interpretation and validation of our methods, we compare these observation-derived results with those obtained from two different COF-evolution models (Elmer/Ice[14] and Specfab[30], detailed in the Supplementary Information Section 2.1 and 2.2, respectively) and to direct ice-core observations from EGRIP in the ice-stream centre[31] and from a shallow ice core (S5) retrieved at the southeastern shear margin (Fig. 2d, e and legend).

## Distribution of horizontal anisotropy in the ice-stream onset
Our analysis of the travel-time differences at airborne radar crosspoints, the beat signature observed in airborne and ground-based radar profiles, and pRES travel-time differences (Fig. 2a–c) consistently show that the depth-averaged difference of eigenvalues in the horizontal plane ($\Delta\lambda$; henceforth called horizontal anisotropy) is generally

high in the ice-stream centre in the upstream part of the NEGIS, suggesting the development of a COF with a pronounced horizontal anisotropy. Downstream of EGRIP, the horizontal anisotropy disappears within a distance of roughly 60 km, particularly evident in the results from the travel-time analysis (Fig. 2a, c), indicating the transition into a vertically symmetric COF (understood as cylindrical symmetry along the vertical axis). The beat-signature analyses show generally weaker horizontal anisotropy (Fig. 2b), which may be related to the averaging nature of the approach and the usage of multi-spectral radar signals subject to slightly dispersive attenuation. The beat signatures are best visible outside the ice stream in the vicinity of the shear margins, where the radar polarisation seems favourably oriented relative to the COF principal axes and where the COF changes over short horizontal distances, leading to power extinction nodes that are dipping downwards, away from the ice stream (see Supplementary Figs. 4–6). The region outside the southeastern shear margin in the centre of the survey area (c″ end of the profile in Fig. 1c) is characterised by heavy folds of deep ice units. These features cause a partial loss of the radar return signal due to steep layer inclination and are, according to our understanding, mainly responsible for the weaker appearance of the beat signature outside the southeastern than the northwestern shear margin in profile c′–c″. Radar profiles crossing the ice stream 20 km further downstream and beyond show beat signatures of similar strength around both shear margins.

Inside the ice stream, the beat signature is more difficult to recognise, presumably because of unfavourable radar-COF alignment and horizontally oriented signatures that are difficult to distinguish from the internal stratigraphy. The data gap in the vicinity of the shear zones is a consequence of strongly folded internal stratigraphy at the shear margins[29] over a width of 5–10 km and throughout the entire ice column but the top ~10% (Fig. 1c). The resulting loss of radar return power related to steeply inclined layers[32] prevents us from obtaining robust results in these regions. Outside the ice stream, the beat-signature analyses show the strongest anisotropic effects near the shear margins, which gradually decrease over distances of less than five ice thicknesses (for better visibility, see Supplementary Figs. 4–6). The travel-time analysis of the pRES measurement outside the shear margin shows clearly notable, but weaker horizontal anisotropy than inside the upstream part of the ice stream, while no notable effects of horizontal anisotropy can be detected with the travel-time analysis of airborne crosspoints in most places outside the ice stream.

The differences in the strength of horizontal anisotropy between the travel-time and beat-signature analyses can be explained by the polarisation angles under which each method performs best: The strongest effect of horizontal anisotropy from travel-time differences is obtained for waves polarised parallel to the COF's principal axes, while the obtained horizontal anisotropy seems weaker for misaligned wave polarisations, and no effect is observable for angles of 45°. The beat signatures, on the other hand, appear strongest for waves polarised around 45° to the COF axes and are non-visible for wave polarisations aligned with the COF (assuming the absence of anisotropic scattering). Both methods, in that sense, provide apparent horizontal anisotropies that are lower-bound estimates and complement each other.

Additional differences between panels a–c in Fig. 2 might arise due to the depth-averaging nature of our results. The travel-time difference analysis from the airborne data is averaged across the manually picked reflections as deep as possible, though the deepest internal reflections were not visible at each crosspoint and the bed reflections are often unclear. The results from the automatic analysis of the beat signature are representative of approximately the upper 1700 m of the ice sheet, while reflections in the pRES measurements were only usable in the upper 1500 m of the ice column. Both the travel-time and the beat-signature analyses, in theory, allow determining the COF variations with depth. However, the vertical resolution of the airborne data

is not high enough to resolve vertical COF variations from travel-time differences and the automatic method for deriving the horizontal anisotropy from beat signatures is averaging the horizontal anisotropy across depth using spectrogram analysis. Depth-varying horizontal COF anisotropy could be derived from the travel-time differences in pRES measurements (Supplementary Fig. 8) and from manually picking the beat signature (Supplementary Figs. 4, 5).

The COF evolution was modelled along flow tubes with the full-Stokes ice-flow model Elmer/Ice[33,34] coupled with a COF-evolution model accounting for lattice rotation[14,16] assuming a steady-state ice-stream geometry. The values of $\Delta\lambda$ vary between 0–0.9, with 95% of values in the range 0–0.67 (Fig. 2d). The very high values ($\Delta\lambda > 0.6$) imply a horizontal single maximum, intermediate values ($\Delta\lambda \approx 0.3$–0.6) generally indicate a vertical girdle, and low values provide little information (vertical single maxima, weak vertical girdles, and isotropy all have weak horizontal anisotropy). COFs are constrained to be universally isotropic close to the surface, and are generally vertical single maxima near the bed, though in some regions, the basal COF is part-way between a vertical girdle and vertical single maximum.

Horizontal shear causes rotation of the horizontal eigenvectors away from the flow direction (Supplementary Fig. 11c), and generally leads to a horizontal single maximum near the ice-stream shear margins. The single maximum is stronger in the northwest margin than the southeast one, despite the northern margin being more diffuse. However, since the COF results from the integrated strain history, this difference is unsurprising; due to residence times in the margin, the total horizontal shear experienced by a flow line is larger in the northwestern shear margin, which is consistent with present-day strain rates observed in the vicinity of EGRIP[35].

Because of stability constraints related to high ice-flow velocities at the model boundaries, the model domain of Elmer/Ice does not cover the entire survey region and does not capture our observations of decreasing horizontal anisotropy in the downstream area. We instead use the Specfab COF evolution model[30] to simulate the COF development for an idealised ice parcel (large enough to statistically represent the COF), initialised with a vertical girdle of the type observed in the EGRIP ice core[31], experiencing the deformation along a downstream flow line. In contrast to Elmer/Ice, Specfab solely simulates the COF development as a response to accumulated strain without being coupled to an ice-flow or temperature model. The COF evolution obtained by this simpler simulation shows a gradual decrease of the horizontal anisotropy, as the COF transitions from a vertical girdle to a vertical single maximum roughly 50 km downstream of EGRIP as a consequence of flow-transverse extension and decreased along-flow acceleration (Fig. 2e). At around 115 km downstream of EGRIP the ice-stream width is nearly constant but flow accelerates again and the COF turns into a vertical girdle again, although in a weaker form than at EGRIP (see Supplementary Figs. 12, 13). Vertical shear is ignored in this simulation since it cannot be inferred from the surface velocities directly, but it could potentially lead to a stronger vertical single maximum COF if vertical shear is non-negligible.

Near the EGRIP drill site, the Elmer/Ice-modelled ($\Delta\lambda \approx 0.66$) and the radar-derived horizontal anisotropy, in particular with the travel-time method applied on pRES and airborne crosspoint data ($\Delta\lambda \approx 0.4$–0.55), agrees well with a vertical girdle observed over large depths in the EGRIP ice core ($\Delta\lambda \approx 0.6$)[31]. The results of our radar-based analyses and the Specfab COF evolution modelling consistently indicate a decreasing horizontal anisotropy in the downstream part of the NEGIS. The Elmer/Ice-modelled horizontal anisotropy ($\Delta\lambda \approx 0.48$) is much smaller than observed at the southeastern shear margin in the S5 shallow core ($\Delta\lambda \approx 0.85$), while in the northwestern shear margin, the simulations indicate the development of a horizontal single maximum comparable to the observations in the S5 core. Due to the lack of signal return power across large depths of the shear margins, the COF

distribution within the shear margins cannot be confirmed independently by radar-based analyses. Outside the NEGIS, the pRES measurement indicates a decreased horizontal anisotropy compared to inside the ice stream, which agrees with the results from the beat-signature analyses and the Elmer/Ice model at the corresponding location.

Both the measured and modelled results suggest that the COF rapidly adjusts to changing strain-rate regimes, confirming findings obtained from COF-evolution modelling studies[5,14]. A vertical girdle develops inside the NEGIS in the upstream and central part of our survey area, where the ice experiences lateral compression, vertical thinning and along-flow extension. Downstream of EGRIP, the flow-transverse extension leads to the removal of lateral compression, while the along-flow extension decreases due to a relatively constant flow velocity. As a result of the cumulative strain related to the slightly divergent flow and the vertical compression, the c-axes rotate into a vertical single maximum over a transition time of approximately 800 years (at modern flow velocities), corresponding to a distance of around 50 km (Supplementary Figs. 12, 13). Further downstream, a weak vertical girdle seems to develop again as a consequence of increasing along-flow extension due to ice-flow acceleration. Increasing flow velocities, convergence, and horizontal simple shear in the shear margins lead to a COF evolution from a putative vertical single maximum outside the NEGIS to a strong horizontal single maximum in the shear margins.

## Soft or hard ice: a question of crystal orientation fabric and deformation type

The local deformation rate of glacier ice depends on its effective viscosity, which is predominantly controlled by temperature and the COF, while other factors such as grain size distribution, impurities, and water content play a notable but minor role[36–39]. Strong COFs have long been known to change the directional viscosity of ice by several orders of magnitude compared to isotropic ice[40,41]. We evaluate the effect of the COF-induced directional viscosities over NEGIS by calculating bulk directional enhancement factors (see Supplementary Information Section 3 and ref. 7 and ref. 30) based on the model- and radar-derived horizontal anisotropy. The enhancement factors are defined as the longitudinal and shear strain rates with respect to the principal COF axes, divided by the expected strain-rate magnitude assuming isotropic ice. An enhancement factor of one is, by definition, identical to the viscosity of isotropic ice under otherwise identical conditions (e.g. temperature and impurity content). Enhancement factors below or above one imply a decrease or increase in susceptibility towards the corresponding deformation and strain orientation, respectively.

The in situ strain-rate enhancements depend on the alignment between the principal strain direction and COF orientation[40]. While the former can partially be estimated from surface velocities, the latter cannot be inferred from direct radar-based observations on large spatial scales, so is limited to model results in our analysis. The enhancement factors presented here are therefore calculated assuming the strain-rate tensor is oriented in alignment with the COF principal frame. In this sense, our method estimates the eigen-enhancements, understood as the maximal effect that COF anisotropy can have on flow under favourable conditions where the strain-rate tensor is aligned with the COF (i.e. an upper-bound estimate). While this is likely to be true for both the ice-stream interior and outside, considerable strain–COF misalignment is expected and modelled in the shear margins (Supplementary Fig. 11). The term 'along-flow' thus represents the direction of the smallest horizontal eigenvalue, which in most places is close to the true flow direction with the exception of the shear margins and their vicinity.

Our radar-based measurements provide information about the horizontal anisotropy, but a full COF description of the second- and fourth-order structure tensors is necessary for the calculation of enhancement factors[30]. Estimating the second-order structure tensor from the horizontal eigenvalue difference requires prior knowledge about the COF type, which we derive from the Elmer/Ice-modelling results and the information from the EGRIP and S5 ice cores. We divide the survey region into three regimes: ice-stream interior, shear margin zone of 4 km width, and the region outside the NEGIS. Inside the ice stream, we assume that the eigenvalue pointing in the flow direction is zero, while outside the NEGIS, it is assumed to be 0.1. The horizontal single-maximum observed in the S5 core and the Elmer/Ice-modelling results lead to the assumption that within the shear zone, one horizontal eigenvalue is of similar size as the vertical eigenvalue. Once the second-order structure tensor is known, the fourth-order structure tensor can be derived by modelling the correlation between the second- and fourth-order spectral coefficients for symmetrical COFs for unconfined compression/extension with Specfab, details of which are described in the Supplementary Information, Section 3.2.

Under the assumptions made, the enhancement factors for along-flow compression/extension (pure shear; $E_{xx}$) derived from the radar data (Fig. 3a, b) inside the ice stream suggest that the ice is 1–2 orders of magnitude harder for pure-shear deformation in flow direction compared to isotropy, and between 5 and 25 times harder compared to the ice outside the NEGIS. Beat-signature and travel-time analysis show a trend towards softer ice downstream of EGRIP ($E_{xx} \approx 0.1$) compared to further upstream ($E_{xx} \approx 0.02$). Enhancement factors based on modelling results (Fig. 3c), which in contrast to those derived from the radar, are not affected by the above-mentioned assumptions, indicate that the ice inside the ice stream is up to ten times harder than isotropic ice for pure-shear deformation. Outside the ice stream, pure-shear enhancement factors from radar methods range from 0.4–0.6, while Elmer/Ice shows an increasing hardening towards the ice stream with enhancement values between 0.4–0.84. These results are consistent with other work showing that differential strain grows during pure-shear deformation, leading to stiffening for further deformation[42]. The shear margins in Fig. 3b stand out as seemingly softer compared to the surrounding ice, which is, however, not confirmed by Elmer/Ice or the S5 core, indicating that the assumptions made to infer the fabric do not hold in the case of the shear zone for the beat-signature anisotropy.

Compared to isotropic ice, the COF makes the ice slightly harder for horizontal shear along flow (simple shear; $E_{xy}$) outside the ice stream, with radar-derived enhancement factors ranging from 0.3–0.7 (travel-time analysis; Fig. 3d) and 0.7–0.9 (beat-signature analysis; Fig. 3e), respectively. Enhancement factors from Elmer/Ice outside the ice stream generally are smaller in the outer flow tubes (0.3–0.5) than in the central ones (0.8–1.4), and increase within the individual model domains towards the shear margins (Fig. 3f). In the ice-stream centre upstream of EGRIP, the ice becomes softer for horizontal shear, although this effect is found to be stronger for the modelled results ($E_{xy} \approx 1.2$–2.0, Fig. 3f) than for the travel-time analysis ($E_{xy} \approx 1.0$–1.4, Fig. 3d), while not observed in the beat-signature analysis (Fig. 3e). Both radar-based flow enhancements suggest that the ice upstream of EGRIP is easier to shear horizontally than downstream (Fig. 3d, e). This is also observed in the central flow line of Elmer/Ice as far as the model domain extends (Fig. 3f).

Radar-based methods provide only limited information on the shear margin viscosity but indicate a slight softening near the shear zones ($E_{xy} \approx 1.0$–1.2 for beat-signature and $E_{xy} \approx 1.3$–1.7 for travel-time analysis). Unsurprisingly, Elmer/Ice shows the strongest shear enhancements in the shear margins, indicating that they are soft for horizontal shear deformation. The modelled shear-margin softening in Fig. 3f is more pronounced in the northwestern shear margin ($E_{xy} \approx 1.7$–2.3) than in the southeastern margin ($E_{xy} \approx 1.1$–1.7) because of the higher residence time in the shear zone of the modelled flow lines. The enhancement factor calculated from the S5 core (displayed in the bottom of Fig. 3), however, suggests that a similarly strong

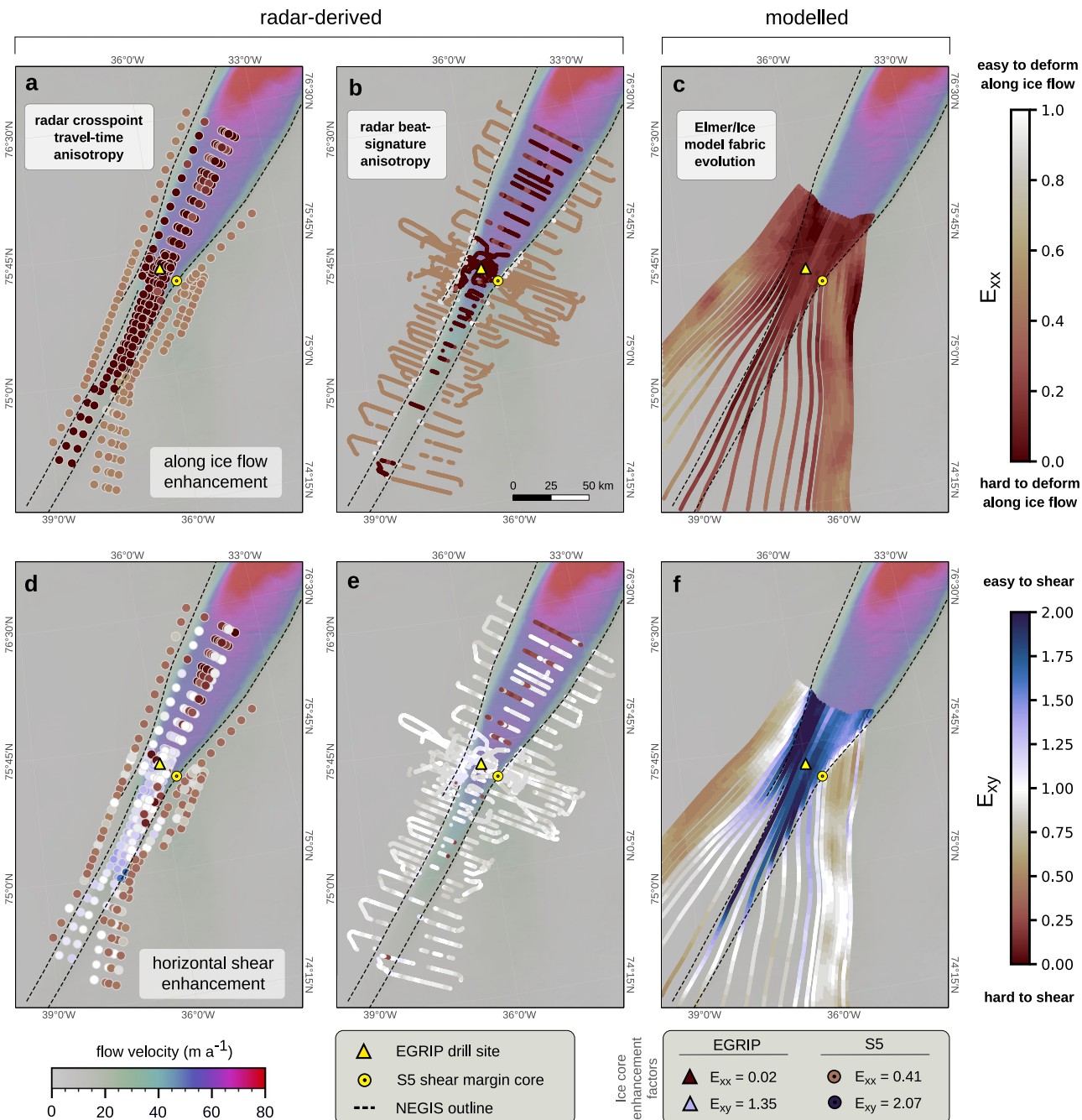

**Fig. 3 | Estimated flow-enhancement factors for radar-derived and modelled crystal orientation fabric (COF) compared to isotropy.** Flow-enhancement factors for along-flow pure-shear compression/extension (**a**–**c**) and horizontal shear deformation (**d**–**f**). Both compressional/extensional and shear enhancement factors are calculated from the COF estimates obtained from travel-time differences (**a**, **d**), beat-signature analysis (**b**, **e**), and Elmer/Ice-flow modelling (**c**, **f**). Note that the enhancement factors displayed in this figure are calculated in the eigenframe (COF coordinate system), since the true orientation relative to the flow direction is unknown, although it can be simulated with Elmer/Ice (see Supplementary Information 2.1 and 3). The term 'along-flow' therefore, refers to the direction of the smallest horizontal eigenvalue. Background in all panels shows satellite-based surface velocities[64].

shear-margin softening of a factor of two could be expected in the southeastern shear margin. However, the modelled COF orientation near the shear margins does not align with the ice-flow direction, so the horizontal shear enhancement along the flow direction and in the modelled direction of the COF principal axes is considerably different (Supplementary Information, Section 3, Fig. 17). Due to missing direct observations of the c-axis orientation in the shear margins, it remains thus unclear what role the COF plays in terms of constraining the ice-stream geometry and/or facilitating rapid ice flow. We also emphasise

that our results based on radar methods are biased by the assumptions upon which our estimate of the second-order structure tensor was derived and the ability of the radar methods to detect the minimum horizontal anisotropy rather than its absolute value. An example of this bias also becomes evident in Fig. 3d at two crosspoints northwest of EGRIP: the radar lines at these crosspoints are not oriented along/perpendicular to the flow direction, so the apparently weaker anisotropy observed from these polarisations results in a different estimated COF type and thus smaller enhancement factors than at nearby

crosspoints. In contrast, the results from Elmer/Ice are unaffected by these biases, which explains the differences between the modelled (panel c, f) and radar-derived (panel a, b and d, e) enhancement factors in Fig. 3.

To illustrate the significance and magnitude of the flow-enhancement for deformation, we compare it to the influence that temperature would have on viscosity by calculating equivalent temperature anomalies for isotropic ice (Supplementary Information, Section 4). We find that a similar along-flow stiffness for pure shear ($E_{xx}$), as suggested by the calculated enhancement factors inside the ice stream, cannot realistically be achieved through temperature, as it would require the ice stream to be -15–30 °C colder than the surrounding ice under the assumption of cold-ice conditions. However, shear-margin softening of a factor of two, as suggested by Elmer/Ice and the S5 core, could also be obtained by a temperature anomaly of 6.6 °C across the shear margins. This estimate is in line with the results of a recent study[43] using a 3D-thermomechanical model, which showed that temperature anomalies of up to 6 °C are plausible in the NEGIS shear margins. Hence, COF and temperature might be equally important for shear margin softening.

## Discussion

The NEGIS is characterised by a strong COF variability over short horizontal distances of a few kilometres in dependence on the flow regime and the ice stream geometry, confirming what was observed in previous studies to some degree too. The horizontal anisotropy obtained from the travel-time analysis near EGRIP agrees with the findings of ref. 28, who used a similar approach to determine profound horizontal anisotropy, which indicates the presence of a vertical girdle COF with c-axes concentrated perpendicular to the flow direction at EGRIP. The increased horizontal anisotropy towards the shear margins observed in the beat-signature analyses agrees with similar studies on Thwaites glacier[25] and Rutford ice stream[27] in Antarctica and is in alignment with direct observations of a horizontal single maximum in a shear-margin core from Priestly glacier[44], similar to what is observed in the S5 core. The good agreement between modelling results and radar-based observations in large parts of the survey area confirms our theoretical understanding of COF development[5,45] and can be regarded as a validation of COF-evolution coupled ice-flow models[14,16,17].

The results obtained with Elmer/Ice suggest that outside the ice stream, the COF is dominated by vertical compression, resulting in a vertical single maximum of increasing strength with depth. Towards the shear margins, horizontal shear becomes dominant and leads to the COF rotation into a horizontal single maximum confirmed by the S5 shear margin core. In the upstream part of the NEGIS, flow acceleration and lateral compression due to flow channelling result in a vertical girdle with a superimposed horizontal single maximum, as observed in the EGRIP ice core. The downstream simulation with Specfab shows that this COF transitions into a vertical single maximum with very weak horizontal anisotropy around 50 km downstream of EGRIP, which is associated with a lateral extension due to slightly divergent flow and constant flow velocity that decreases the flow-parallel extension component. Flow acceleration and a near-constant ice-stream width further downstream (around 116 km from EGRIP) lead to a further COF transformation into a vertical girdle, although in a weaker form than upstream.

The influence of the COF on the effective viscosity depends on the deformation type and direction (Fig. 4). We find that in the upstream part of NEGIS, the COF leads to considerable stiffening of the ice stream for pure-shear deformation in the flow direction. While several mechanisms are known to make ice softer, crystal orientation is, to our knowledge, the only factor that can harden ice by the order of magnitude suggested by our results. Our direct radar-based observations do not allow us to determine the COF type and its effect on the effective viscosity without ambiguities because of vertical symmetry

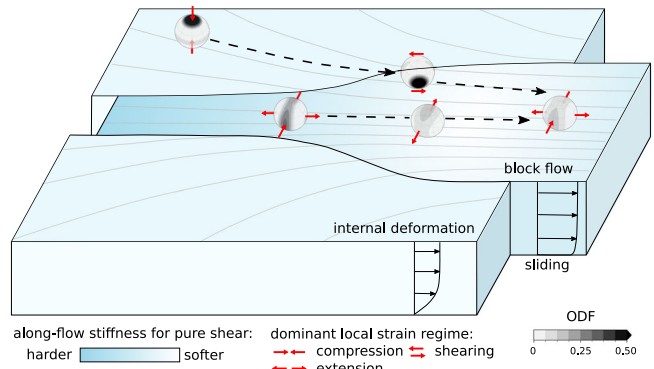

**Fig. 4 | Summary of flow mechanics and evolution of the crystal orientation fabric (COF) in the ice-stream onset region.** The COF distribution in the Northeast Greenland ice stream (NEGIS) is a result of the deformation history. The dominant vertical compression outside the ice stream leads to a vertical single maximum, which rotates into the horizontal plane towards the ice-stream margins, where horizontal shear is the dominant deformation mechanism. In the upstream part of the NEGIS, ice-flow channelling and along-flow acceleration create a vertical girdle with a superimposed horizontal single maximum. Downstream of the East Greenland ice-core project (EGRIP), divergent flow and stagnant flow velocities lead to a reversed deformation, so c-axes rotate back into vertical symmetry. By the downstream end of the survey region, stagnant ice-stream width and increased flow velocities again cause along-flow extension and the transition into a girdle-type COF. The COF affects the ice viscosity, e.g. leading to considerable stiffening for pure-shear deformation in parts of the ice stream.

(e.g. isotropic and vertical single maximum COFs would have the same effect on nadir-pointed radar measurements), so the enhancement factors are considerably affected by assumptions based on COF evolution modelling.

Numerous observations demonstrate that calving events can lead to increased ice-flow velocities far inland[46–48], and in particular in ice streams with low basal friction[49]. Calving and similar perturbations at the glacier front or at the bed, propagate upstream through geometrical changes in the flow field or through direct transmission of membrane stresses (i.e. longitudinal stresses). A previous study[50] found that disturbances on decadal or subdecadal forcing periods propagate upstream through processes dominated by pure shear and that the decay length of the perturbations is longer for harder ice. A simple calculation of the characteristic time for viscous response (Supplementary Information, Section 5) suggests that the reaction time for pure shear is approximately ten times longer for ice with $E_{xx} = 0.1$ in comparison to isotropic ice. Our findings are, therefore, potentially relevant for understanding how modern changes at the ice-sheet margin will affect long-term mass loss, as they imply that surge events or changes in basal water pressure, for instance, could propagate farther upstream and result in a delayed response in ice with COFs that cause along-flow stiffening. While it is unlikely that perturbations at the ice-sheet margins propagate so far inland to affect our survey area[51], glaciers and ice streams in near-coast regions could potentially be more sensitive towards external disturbances due to COF-induced viscosities than commonly assumed, though the extent to which this affects ice-sheet dynamics remains to be examined in future studies. Recent findings[52] showed that the differences in simulated surface velocities between full-Stokes models and higher-order models using e.g. the Blatter-Pattyn approximation is much larger for ice with higher viscosity in comparison to softer ice. To the extent that we find stiffening in the along-flow direction, our results indicate that full-Stokes models are more suitable for highly dynamic areas such as ice streams.

Although our results suggest that the ice-stream margins are soft for horizontal shear, the role they play in maintaining the ice stream remains unclear. The horizontal single maximum observed in the S5

core and simulated by Elmer/Ice could facilitate horizontal shear by up to a factor of two. A similar enhancement could, however, be obtained from strong shear heating, as well as by the presence of temperate ice or water in the shear margins[53]. Both temperature and the COF likely play a role in the shear-margin viscosity; however, we cannot determine whether the COF is oriented in a preferable way to facilitate shear deformation in the NEGIS shear margins.

## Methods

The orientation of ice crystals is commonly represented as a second-order orientation tensor, $\mathbf{a}^{(2)}$, describing the density distribution of c-axes orientations: its eigenvectors, $\mathbf{a}_i$, and the corresponding eigenvalues, $\lambda_i$, describe the orientation and length of the three principal axes[54–56]. By convention, the three eigenvalues are defined as $\lambda_1 \leq \lambda_2 \leq \lambda_3$ and $\lambda_1 + \lambda_2 + \lambda_3 = 1$. We derived the horizontal COF anisotropy ($\Delta\lambda$) from ground-based and airborne radar profiles, as well as from polarimetric pRES measurements at distinct locations, assuming that one of the principal eigenvectors is vertically oriented. These results are complemented and validated by COF evolution models, as well as observations from the EGRIP and S5 deep and shallow ice cores, respectively. Flow-enhancement factors were calculated from the estimated COF. Details on each method, validation and uncertainties are presented and further discussed in the Supplementary Information: The analytical methods of travel-time and beat-signature analyses are described in Section 1, model details and the performed simulation to obtain the modelling results are described in Section 2, the calculation of enhancement factors and the underlying assumptions are detailed in Section 3, temperature difference estimates corresponding to enhancements caused by COF are further described in Section 4, and an estimate of the characteristic time is shown in Section 5.

## Data

We used an extensive set of airborne radar sounding data (EGRIP-NOR-2018 survey) which was recorded in 2018 over the onset region of NEGIS and its shear margins[29]. The survey was conducted with a Multichannel Coherent Radar Depth Sounder (MCoRDS 5) mounted on the Polar 6 BT-67 aeroplane[57]. The radar system consists of eight antenna elements, each functioning as a transmitter and a receiver using a transmit–receive switch. The antennas were oriented such that the E-field polarisation is parallel to the flight direction (HH polarisation), and the transmitting wave beam was pointed towards the nadir. The profiles were recorded using linear frequency-modulated chirps in the frequency band of 180–210 MHz at a pulse repetition frequency of 10 kHz, and the received signals were sampled at a frequency of 1.6 GHz. The aeroplane flew at an approximate altitude of 360 m above ground and a velocity of 260 km h$^{-1}$. Processing was performed with the post-processing CReSIS Toolbox[58] and includes pulse compression using a Tukey time–domain weighting and filtering with a Hanning window. We refer to ref. 29 for further details.

The airborne survey was complemented with a dedicated ground-based UHF radar profile (see Supplementary Fig. 4), recorded with an 8-element transmit and receive system in T-configuration and a mean frequency of 750 MHz with a bandwidth of 300 MHz. This system provides a higher horizontal and vertical resolution than the airborne data, but covers less area. Further system details and the data processing are described in ref. 59.

In addition, we used polarimetric measurements with a ground-based phase-sensitive radio-echo sounder (pRES)[60,61], which allows to determine vertical displacements of repeated measurements with accuracy in millimetres. Between 2017 and 2019, we performed six polarimetric pRES measurements inside and outside the ice stream within 85 km around EGRIP. At each site, besides the HH-polarised measurement, a measurement with VV-polarisation was carried out by horizontally rotating the transmitting and the receiving antennas by 90°. During each measurement, the pRES transmitted 100 chirps, each

ranging from 200 to 400 MHz over a period of 1 s. The received signal was sampled at 40 kHz. For data processing, we followed ref. 60 and ref. 62 in order to get amplitude and phase profiles as a function of two-way travel time.

Observational COF data were retrieved from two ice cores: the deep EGRIP ice core and a close-by shallow core (around 14 km from EGRIP) in the southeastern shear margin of NEGIS called S5. The preparation and measurement of the samples from both ice cores follow the same procedures: All samples were analysed from thin sections that were cut vertically to the ice-core axis and have dimensions of about $90 \times 70 \times 0.3$ mm$^3$. The sample surfaces were carefully polished with a microtome knife in the EGRIP trench at –18 °C. After one hour of controlled sublimation, c-axes were measured with an automated COF analyser by Russel-Head Instruments (FA G50). The data were background corrected before processing, and the COF was derived via digital image processing. Here, we used the COF measurements to validate our geophysical observations and modelling results. The COF information from the EGRIP ice core stems from depths of 1830, 2024, and 2087 m, as published in ref. 31, and shows a vertical girdle with a superimposed horizontal single maximum. Three samples at a depth of 68 m in the S5 shallow core show a strong horizontal single maximum COF.

### Travel-time difference and beat-signature analyses

Radar waves are decomposed into ordinary ($e_x$) and extraordinary ($e_y$) components when travelling through polycrystalline ice with bulk anisotropic dielectric properties. Varying wave speeds for waves polarised in different directions result in a two-way travel-time difference $\Delta t$ when reflected off the same reflector at depth and received at the surface. The value of $\Delta t$ at a certain depth depends on the strength of the horizontal anisotropy. At points where two radar waves are polarised orthogonal to each other (i.e. crosspoints of the airborne radar lines and the pRES measurements repeated with orthogonal antenna orientation), the travel time of reflections can be exploited to derive the directional depth-average dielectric permittivities and the degree of horizontal anisotropy. For the airborne data, we manually picked the arrival times of internal reflections in crossing radar profiles. The depth of these reflectors varied from 450 to 3000 m and was determined with a relative permittivity profile derived from Dielectric Profiling (DEP) measurements on the EGRIP ice core. The depth-averaged apparent horizontal anisotropy was calculated from the slopes of a linear regression through the picked arrival times with depth (Supplementary Equation 7–10 and Supplementary Information, Section 1.1).

A similar approach was applied to the pRES data, with the main difference being that its high vertical resolution and phase sensitivity allows for determining the depth variance of horizontal anisotropy (Supplementary Information, Section 1.3). The analysis of travel-time differences provides an estimate of the apparent horizontal anisotropy, which is close to the absolute value for antenna polarisations that are parallel to the COF's principal axes but is smaller if the antennas are not aligned with the COF. For antennas rotated 45° from the COF principal axes, no effect of horizontal anisotropy can be detected even if it is present.

For a transmitted wave whose polarisation plane is not aligned with one of the horizontal anisotropy axes, the electric field can be regarded as the superposition of the two wave components, $e_x$ and $e_y$. In horizontally anisotropic ice, the difference in the wave speed between these two wave components leads to a phase shift and hence a polarisation rotation, causing extinction nodes in radar return power[23,63]. In the absence of anisotropic scattering, these nodes are most pronounced when the transmitted wave is polarised at an angle of 45° from the COF principal axes, and the phase differences are proportional to the radar wave frequency and the degree of horizontal anisotropy[23].

Here, we manually picked the beat signatures in selected airborne and ground-based radargrams to determine the horizontal COF anisotropy (closely following the technique by ref. 25). Additionally, we developed an automated method that was applied to the entire airborne data set. Due to the automatisation, the resulting horizontal anisotropy of the beat-signature analysis is a depth average, representing approximately the top 1700 m of the ice sheet. As for the travel-time difference analysis, the beat signature is only visible for certain antenna polarisations in relation to the COF principal axes. Consequently, The inability of these methods to detect horizontal anisotropy at particular polarizations does not prove the absence thereof. Further details on the manual and automatic methods can be found in the Supplementary Information Section 1.2.

### Modelling the evolution of the crystal orientation fabric

We used the ice-flow model Elmer/Ice[33,34] to simulate the COF evolution in the catchment area of NEGIS, assuming that the COF evolves solely by lattice rotation, while recrystallisation processes are ignored. The model couples the processes of COF evolution, ice flow and heat flow[14], and was solved along 31 flow tubes spanning the full ice thickness.

The width of the flow tubes was obtained by tracing particle paths upstream in the measured InSAR velocities[64], starting from a flow tube half-width of 2.5 times the ice thickness (total width of 12.5 km) near the EGRIP core site. The flow tubes are wide enough to average out some of the effects of small-scale topography that can detrimentally affect flow line models[65], but overlap with each other at some points (Supplementary Fig. 10). Note that convergence towards the ice stream leads to a significant narrowing of the central flow tubes, while outer model domains experience little effect from the extra half dimension. The ice surface and bed elevations defining the model geometry were determined using the average values in the across-flow directions from ArcticDEM[66] and Bedmachine v3[67], respectively. In order to perform all calculations in ice-equivalent thickness, surface elevations were modified by subtracting 18 m everywhere to account for the firn-air content as estimated from RACMO 2.3.1[68]. Each model domain used a triangular mesh with 100-m vertical and 250-m horizontal resolution. For each flow tube, we ran a 5-kyr transient simulation, after which changes in COF in the areas of interest were negligible. Initial conditions were reasonable guesses for COF and temperature, but simulations were long enough that there is no sensitivity to the exact choice. Time stepping used a second-order backwards difference with variable time steps of 0.01–1.0 years.

Though this is arguably the most advanced coupled model of COF evolution and flow, it still has a number of shortcomings, particularly for dynamic areas. First, the COF evolution considers only lattice rotation; a number of studies suggest a role for dynamic recrystallisation in the evolution of COFs in warm ice, and even in colder ice when stress is high (see ref. 4 and references therein). While the full stress state is included by the parameterised half dimension, the model is not fully three-dimensional. As a result, the across-flow resolution is very coarse (effectively 2.5 km), and the horizontal shear stresses must be imposed. This requires smoothing of remotely sensed velocity products to obtain reasonable strain rates, which may degrade sharp gradients in the COF. While the model assumes isotropic ice at the surface, ice cores often show anisotropic COFs near the surface[31,69]. Despite these limitations, the model matches the pattern of COF inferred from radar and measured by ice cores well enough (i.e. within the uncertainties of the methods) to provide a reasonable estimate of the COF between data points. Further details of the model and our application in the NEGIS can be found in ref. 14 and Section 2.1 in the Supplementary Information.

Due to the loss of numerical stability associated with high ice-flow velocities, the domain modelled with Elmer/Ice only extends 40 km downstream of EGRIP. To explore the COF evolution in the downstream part of NEGIS, we used the spectral COF model for polycrystalline materials 'Specfab' by ref. 30. In contrast to Elmer/Ice, Specfab is not an ice-flow model but solely simulates the COF evolution. The model is a kinematic model in the sense that c-axes rotate in response to the velocity gradient field (apparent strain-induced rotation of c-axes), and not stress or temperature.

We considered an ice parcel which is statistically representative of the COF seeded at EGRIP with an initial COF similar to the average girdle-type measured in the ice core[31] and let it travel downstream along a flow line obtained from the surface velocities by ref. 64. The Lagrangian COF update of the parcel for incremental steps of 1 year is given by strain-rate and spin tensors derived from the surface velocities in the corresponding flow line segment. The end of the simulation corresponds to a distance of ~120 km from EGRIP and an advection time of 2000 years. Further details of the model and our application can be found in Section 2.2 of the Supplementary Information and in ref. 30.

### Flow-enhancement factors

To estimate the effect of the COF on directional ice viscosities, we calculated the directional enhancement factors. The enhancement factors are defined as the strain-rate ratio between anisotropic and isotropic ice, whereby the anisotropic rheology is estimated by averaging a transversely isotropic monocrystal rheology over all grain orientations in a polycrystal[7,30]. While our radar-based COF analyses only allow inferring the difference in horizontal eigenvalues ($\Delta\lambda$), the required full COF estimate can be obtained by applying a few assumptions about the expected COF types inside/outside the ice stream and in the vicinity of the shear margin. These assumptions (detailed in the Supplementary Information, Section 3.1) are based on observations in ice cores and results obtained from COF evolution models.

Besides the COF, the temperature is another major factor controlling the ice viscosity. For comparison, we used the calculated enhancement factors to estimate the temperature difference ($\Delta T$, relative to a reference temperature of −20 °C) that would be required to obtain ice softening/hardening equivalent to that suggested by the COF. Thereby, $\Delta T$ is derived from the formula of temperature-dependent enhancement factors assuming isotropic ice and setting the enhancement factors to the values obtained from COF effects. For further details, see Supplementary Information, Sections 3 and 4.

## Data availability

The depth-average horizontal anisotropy distribution and the estimated flow-enhancement factors generated in this study have been deposited in the ERDA database: https://doi.org/10.17894/ucph. ed9a1a1f-d6e6-41d5-894c-7e526f75fdd7[70]. The same archive also contains the raw data of picked reflections used for the crosspoint travel-time analysis, the processing files from the beat-signature analysis and the data files to reproduce Figs. 1–3. The raw airborne radar profiles are available from the World Data Centre PANGAEA: https://doi.org/10. 1594/PANGAEA.928569[71]. An example data set for one of the pRES measurements is described by ref. 72 and publicly available from the World Data Centre PANGAEA: https://doi.org/10.1594/PANGAEA. 951267[73]. The ground-based UHF profile is deposited in the ERDA database: https://doi.org/10.17894/ucph.4f771760-fb08-4a97-9e48-818b8c7601d8[74].

## Code availability

The Specfab model is publicly available on the github repository: https://github.com/nicholasmr/specfab. The slightly modified Elmer/Ice[34] model code is the same as used by ref. 14 and is available at: https://github.com/dlilien/elmerfem/tree/release-8.4.1[75]. Scripts for the beat-signature analysis (https://github.com/oeisen/radar-beats[76]), crosspoint travel-time analysis (https://github.com/tamaragerber/NEGIS_traveltime_analysis[77]), downstream COF

simulation (https://github.com/tamaragerber/Specfab_COFevolution_EGRIPflowline[78]) and the calculation of enhancement factors (https://github.com/tamaragerber/NEGIS_enhancementFactors[79]) are available in the corresponding Github repositories.

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

## Acknowledgements

This research was undertaken, in part, thanks to funding from the Canada Excellence Research Chairs Programme and has been

financially supported by the Villum Investigator Project IceFlow (Grant No. 16572 to D.D.-J.). Radar development was further supported by funding from the University of Alabama. EGRIP is directed and organised by the Centre for Ice and Climate at the Niels Bohr Institute, University of Copenhagen. It is supported by funding agencies and institutions in Denmark (A. P. Møller Foundation, University of Copenhagen), the USA (US National Science Foundation, Office of Polar Programmes), Germany (Alfred Wegener Institute, Helmholtz Centre for Polar and Marine Research), Japan (National Institute of Polar Research and Arctic Challenge for Sustainability), Norway (the University of Bergen and Trond Mohn Foundation), Switzerland (Swiss National Science Foundation), France (French Polar Institute Paul-Émile Victor, Institute for Geosciences and Environmental Research), Canada (University of Manitoba), and China (Chinese Academy of Sciences and Beijing Normal University). S.F. received funding from the German Academic Exchange Service (DAAD): Forschungsstipendien für promovierte Nachwuchswissenschaftlerinnen und -wissenschaftler. M.R.E. was supported by a DFG Emmy Noether grant (grant no. DR 822/3-1). The authors would like to thank the EGRIP logistic support and field personnel, the US Air National Guard flights provided by the US National Science Foundation as well as Ken Borek aircraft crew.

## Author contributions

O.E. and T.A.G. designed and coordinated this study. T.A.G. evaluated the airborne radar crosspoints and developed the approach used here to derive horizontal anisotropy from travel-time differences. O.E. developed the spectrogram analysis, T.J.Y. developed the node-line analysis, F.V.-D., T.J.Y. and O.E. analysed the beat signatures. D.A.L. performed the COF evolution simulations with Elmer/Ice. M.R.E. simulated the beat frequency modulation under the supervision of R.D. O.Z. and A.H. performed and analysed the pRES measurements. N.S. and I.W. analyzed the COF of the EGRIP and S5 cores. N.M.R. and T.A.G. designed and performed the calculations of the enhancement factors and equivalent temperatures with help from A.G. and C.S.H. N.M.R. and T.A.G. simulated the downstream COF evolution with the Specfab model. O.E. and D.J. were PI and Co-PI of the airborne radar campaign, SF and D.J. processed the data with support from V.H. and J.P. D.S., V.H., C.O., S.P.G., H.M., D.J. and O.E. designed, coordinated resp. collected and processed the ground-based radar survey. S.F. and T.A.G. made the main figures of the article. T.A.G., O.E., D.A.L., N.M.R., T.J.Y., M.R.E. and S.F. wrote the manuscript and Supplementary Information with contributions and comments from all remaining authors.

## Competing interests

The authors declare no competing interests.
