## [Peer Review File · Nature Communications]

Crystal orientation fabric anisotropy causes directional hardening of the Northeast Greenland Ice StreamREVIEWER COMMENTS

Reviewer #1 (Remarks to the Author):

Review comments on "Crystal fabric anisotropy causes directional hardening of the Northeast Greenland Ice Stream" by Gerber et al.

1. General comments:

This paper investigates ice crystal orientations in the vicinity of a deep ice core drilling site in northeastern Greenland. The analysis is based on airborne and ground based ice radar surveys, which are supplemented by numerical experiments using two different numerical models as well as ice core data. The findings are discussed in connection with ice mechanical property and its influence on the ice sheet dynamics. The authors argue that the ice anisotropy revealed by the study is potentially important for the future evolution of the ice sheet.

I congratulate the authors who synthesized a variety of measurements, analyses, and simulations to improve our understanding of ice crystal orientation in the Greenland ice sheet. The importance of the subject has been recognized, but existing knowledge is limited particularly for fast flowing regions of ice sheets. To my knowledge, this is the most comprehensive study of that kind performed as a combination of state of the art ice radar survey, numerical modeling, and ice core analysis. The findings are nicely illustrated in Figure 4, which provides important implications for the ice sheet dynamics. The text is well written. The main text is easy to understand for a broad range of readers, whereas supplemental material explains details for those specialized in the study field.

My only concern is the style of the presentation, i.e. very compact text with only four diagrams accompanied by a large volume of supplementary information consisted of 20 figures and 52 equations. In my opinion, the most important and novel part of the study is the data analysis and numerical modeling, rather than a relatively speculative implication of the result. It was hard for me to follow the main text until I spent longer time for the supplementary information. Therefore, I hope the paper is presented in a more ordinal way with details of the methodology and results. I believe Nature Communications allows space for that.

Overall, I find the study reports important results obtained by a comprehensive and innovative methodology. My suggestion is to move some portion of the supplementary information to the main text, so that the results are easier to understand and the central part of the study is more exposed to readers.

2. Major concerns

(1) Presentation in the main text

I understand the importance of the study, but I think the implication of the results is not so simple. I am also afraid that the study results do not carry critical and/or immediate impact on the research field. Rather, the careful analysis of the data that obtained by a huge effort in the field provides the basis of the research on the ice sheet dynamics. The main text nicely explains the overview of the study, but excludes necessary and valuable details of methodology and results. I would like to encourage the authors to include text, figures and equations, that are currently presented as supplementary information. For example, Figure 4 in the supplementary information is very important because it explains how vertically mean anisotropy was obtained from the airborne radar data. Moreover, this plot suggests shortcomings of taking the depth-averaged anisotropy. Anisotropy is apparently greater near the bed, which is important and worthwhile for discussion. Similarly, Figure 9 in the supplementary information is essential to describe anisotropy obtained from the phase sensitive radar data. Figure 13 is important as well, because it forms the basis of the enhancement factor calculation. I also like to see Figure 3 (in the supplementary information) in the main text, which shows all the data obtained at the cross sections of the radar profiles (I understand that Figure 2a in the main text shows only a part of the data).

It is difficult to say which to be included in the main text, and it is of course a matter of choice by the authors. I wish the authors present more about what you did and obtained, rather than rounding off the details and focusing on relatively uncertain implications.

(2) Depth-averaged horizontal anisotropy

I was rather surprised at the procedure to average the anisotropy over the full ice thickness. As clearly seen in Figure 4c in the supplementary information, development of the fabrics is expected in the deeper region. I wonder if it is appropriate to smooth the vertical distribution of the anisotropy. It is actually possible to discuss vertical distribution of the anisotropy using the data (Figure 4c and 9).

(3) Comparison with previous studies

Ice fabrics in ice sheets have been studied by various methods in Antarctica as well as in Greenland. Comparisons with such previous studies are completely missing in the article. How do you compare your results and methodology with those reported in Greenland and Antarctica? It is important to demonstrate how you improve our existing knowledge of ice fabrics and its influence on the ice sheet dynamics.

3. Specific comments

Abstract: The abstract is easy to read for general readers, but I think it is too general as an abstract of a scientific paper. Only two sentences in lines 9-13 are directly related to the finding of the study. The last sentence (application to extraterrestrial ice and other materials) is not very much relevant with the study. I suggest the authors to describe more about the study results, namely ice fabrics and its distributions that you revealed by radar, modeling and the ice cores.

Line 4: The definition of "fabric" is inconsistent with that given in 19-20.

Line 88-90: It is very difficult to see significant variations in the anisotropy obtained by beat (Figure 2b). Do you really think it is consistent with the result from the airborne radar?

Line 93-95: Which data show this trend? Figure 2a shows no data out side of the shear margins, whereas I see no significant variations in Figure 2b.

Line 206-223: Do you think a calving event at the glacier front affects the ice dynamics >500 km inland? I suppose chain reactions, such as "acceleration >> thinning >> steepening >> acceleration >> ..." may propagate inland, but not a single calving event in a decadal or subdecadal time scale. Further, I believe such dynamical response of an ice stream is more affected by horizontal shearing near the side margins as well as vertical shearing near the bed. Thus, numerical experiments are required to investigate the response of the ice dynamics to downstream perturbation. I find the discussion in this paragraph is exaggerated and misleading.

Supplementary information: Texts are very well written. However, some portions should go into the main text, and some others are not necessary. For example, Line 7-23 are nice introduction of ice fabrics, but only if this is a dissertation or a text book. I think they are too lengthy and out of the scope of paper supplement. I also suggest to reduce equations, which are appeared in the citing articles.

Line 323-324 in Supplementary Information: Can you clarify the meaning of "close-by shallow core" and "samples are vertical to the ice-core axis"?

Line 472-483 in Supplementary Information: I think this is an important point which should be discussed in the main text.

Figure 5 in Supplementary Information: "Lower panels" are mentioned in the caption, but not appeared in the figure. Mixed up with Figure 6?

Line 626-634 in Supplementary Information: Discussion here is rather qualitative. It is not clear what is the conclusion of the calculation.

Line 639-640 in Supplementary Information: Why "larger viscosity" results in "shorter characteristic time"? It contradicts to Equation (52).

Reviewer #2 (Remarks to the Author):

Overview

I believe this paper provides insight into fabric development and viscous anisotropy in real-world locations. It uses a novel combination of a wide range of modelling and observational data sets to provide this. In this review, I have focused on the fabric and fabric evolution methods as that is my area of expertise. The text is well written, clear and easy to read throughout. However, in places the manuscript could benefit from being clearer about the assumptions it uses and provide more explanation where results don't agree. I also have some questions about the isotropic fabric modelled downstream of EGRIP. If these are addressed, I believe the manuscript is suitable for publication and will be a very useful contribution to the field.

Main Comments

1. Explanation needs to be given over which depths these fabrics are valid. I would expect them to be valid to a reasonable depth into the ice sheet, but it needs to be explained in the text. How deep does the radar data go? This also links in with the expected fabric downstream as there is unlikely to be significant vertical shear in the top 50% of the ice stream. In general, I think it would be clearer to refer to these results as applying to 'fabrics in the top xx% of the ice stream' throughout the manuscript, as the fabrics very close to the base are likely to be different.
2. For the downstream fabric, I'm confused as to what process causes the fabric to return to isotropic. If a diffusional or regularization process is causing this, is the magnitude of it realistic? [1] finds a pre-factor of approx. $0.3 \sqrt{D:D/2}$ in front of the diffusional term is valid for experimental strain-rates (where D is the strain rate tensor). This value is likely to be lower still at the lower strain-rates at EGRIP. Is this isotropic fabric produced by too much 'regularization' in the model?
3. Supplement line 504: 'In addition, the spin is assumed to vanish'. This may be the case, but more justification should be provided. If the along flow gradients are small, then even a small amount of curvature could make the vorticity non-negligible. This should be tested, especially as the model can easily accommodate this and the satellite data available contains this information.
4. Line 172 and Fig 3. Why the disagreement between model and radar over easy-to-shear. I don't think you can say 'Both radar- and model-based flow enhancement suggest that the ice upstream of EGRIP is easier to shear horizontally than downstream' as the radar data seems to show no change from isotropic upstream or downstream. There should be some discussion in the text about why the Elmer/Ice results diverge so much from radar data here. This is especially interesting as it appears in Fig 2. That Elmer/Ice and the radar data agree for fabric predictions.

Minor comments

Generally, in the manuscript you use COF or fabric interchangeably, I think it would be clearer to choose one and stick with it.

For someone who is not an expert on RES, I would appreciate a bit more explanation of how this process works. For example, I had to look up what birefringent was.

Line 59: Here I was a bit confused initially what kind of anisotropy you are referring to? Horizontal anisotropy or viscous anisotropy which comes up later in the manuscript. I think throughout it would be good to always be clear what kind of anisotropy you are referring to.

Line 129: Similarly, to my main comment 2, you say the stress relaxes, but what deformation causes the c-axes to rotate to vertical symmetry, if there was no deformation then the c-axes would not rotate.

Line 161: A sentence or two on these assumptions is needed here, rather than the reader having to refer to the supplement, especially as they are quite strong assumptions. Having looked at the supplement, I agree that they are reasonable, but they should be discussed to some extent.

Fig 3. Similarly to above, I don't think it is correct to call a,b,c,d 'measured' enhancement factors, as it relies on some guesses for the eigenvalues. A better subtitle could be 'measurements + approximations' or words to that effect.

Line 175 - should this be a reference to Fig 3.f as well?

Line 179: flow-direction should not be hyphenated.

Line 193: do you have a reference for the average temperature being -20C?

Line 300 and in the supplement: There should be more discussion on the equivalence/difference between the fabric evolution models, and on the Specfab model, As I had to read [2], and the github page to understand. Does Specfab include a diffusional term?

I hope the authors and the editor find these comments useful

Best wishes,
Daniel Richards

References

[1] D. H. M. Richards, S. S. Pegler, S. Piazzolo, and O. G. Harlen, 'The evolution of ice fabrics: A continuum modelling approach validated against laboratory experiments', *Earth Planet. Sci. Lett.*, vol. 556, p. 116718, Feb. 2021, doi: 10.1016/j.epsl.2020.116718.

[2] N. M. Rathmann, C. S. Hvidberg, A. Grinsted, D. A. Lilien, and D. Dahl-Jensen, 'Effect of an orientation-dependent non-linear grain fluidity on bulk directional enhancement factors', *J. Glaciol.*, vol. 67, no. 263, pp. 569–575, 2021, doi: 10.1017/jog.2020.117.

Reviewer #3 (Remarks to the Author):

This paper investigates how the ice fabric translates to mechanical anisotropy of ice in the onset region of the NEGIS, Greenland. The study uses a set of independent techniques comprising of radar sounding, ice core analysis and numerical modelling. The noteworthy results are the maps of measured/modelled flow enhancement factors for pure shear compression/extension and horizontal shear deformation in the NEGIS

region. These are depicted in Fig. 3 of the main manuscript. Under assumptions made by the authors, their results show that the ice in the upstream portion of NEGIS is an order of magnitude harder than isotropic ice further downstream. Also, the upstream ice is around three times harder than ice outside the ice stream. Data is also presented on depth-averaged fabric distribution and flow enhancement factors.

The work is significant to the radioglaciology community and it builds upon established literature that have used some of the present techniques in other ice stream areas. The relevant references for such have been appropriately cited. To the best of my knowledge, the work presented in this paper is the first for this dynamic region of Greenland.

The claim in the abstract and conclusion on transferring the methodology to other planets and geologic materials requires additional supporting evidence. It is currently not a well-justified claim not least because the radar and coring technologies for extra-terrestrial ice characterisation will differ considerably to those applied in Greenland in this paper. At the very least, some discussion on how the authors expect their technique can transfer to extra-terrestrial ice characterisation and what the limitations are could strengthen their claim. Clarification should also be made for the ice inside and outside the ice stream: in the abstract it is claimed to be two times harder, but the main text suggests three times harder inside.

The data analysis and interpretation on the whole are sound. I would have like to see more detail in the comparison with the different methods used i.e. the radar sounding, ice flow modelling and ice core analysis. The presentation of the data analysis in the form of 2D maps does not make clear how the different techniques support each other and to what extent.

I consider the authors' methodology is sound. Some of plots such as the radargram are missing key scalebars which are normally expected to be included. Another minor detail I found to be missing is the explicit GPS coordinates of the cross-point locations for part of the work to be reproduced.

In summary, I enjoyed reading your paper and its supplementary. Please find attached some minor corrections/suggestions that may help improve your manuscript. In general, I found it well written but a little disjointed in places especially in the supplementary, which currently reads as if different sections have been written by different authors but without an overarching edit that joins them up consistently. As you will see in my comments, there are many areas where I felt specific quantification are lacking in the expressions used to describe your results and observations. These could be tightened up.

It is impressive that you have been able to use multiple independent techniques to arrive at the key results for this region of Greenland. While the maps in Fig. 3 are useful and key, I believe that an additional simple 2D plot directly comparing the results from the different radars, ice flow modelling and ice core analyses would be more meaningful. For instance, an X-Y plot of the Δ_{λ} versus flow direction along the centre of the ice stream. Such a plot could overlay the results derived from the radars, modelling and ice cores – which would greatly strengthen the support of each other. It is hard, as a potential reader, to see/compare in any quantitative detail the differences in results from the different techniques from the existing 2D maps produced.

I would also recommend some additional discussion on the asymmetry, if any, of the fabric observed on both shear margins. From the Fig. 1, the extinction node lines are visible in the radargram (highlighted in box d) but on the opposite shear margin it is less pronounced.

Author's response to review # 1

January 21, 2023

Dear reviewer,

Thank you for taking the time to read the manuscript and its supplement. We appreciate your feedback and find it helpful to improve the impact of our work. Please find the detailed response to your comments below.

On behalf of all co-authors,

Tamara Gerber

1 General comments

This paper investigates ice crystal orientations in the vicinity of a deep ice core drilling site in northeastern Greenland. The analysis is based on airborne and ground based ice radar surveys, which are supplemented by numerical experiments using two different numerical models as well as ice core data. The findings are discussed in connection with ice mechanical property and its influence on the ice sheet dynamics. The authors argue that the ice anisotropy revealed by the study is potentially important for the future evolution of the ice sheet. I congratulate the authors who synthesized a variety of measurements, analyses, and simulations to improve our understanding of ice crystal orientation in the Greenland ice sheet. The importance of the subject has been recognized, but existing knowledge is limited particularly for fast flowing regions of ice sheets. To my knowledge, this is the most comprehensive study of that kind performed as a combination of state of the art ice radar survey, numerical modeling, and ice core analysis. The findings are nicely illustrated in Figure 4, which provides important implications for the ice sheet dynamics. The text is well written. The main text is easy to understand for a broad range of readers, whereas supplemental material explains details for those specialized in the study field. My only concern is the style of the presentation, i.e. very compact text with only four diagrams accompanied by a large volume of supplementary information consisted of 20 figures and 52 equations. In my opinion, the most important and novel part of the study is the data analysis and numerical modeling, rather than a relatively speculative implication of the result. It was hard for me to follow the main text until I spent longer time for the supplementary information. Therefore, I hope the paper is presented in a more ordinal way with details of the methodology and results. I believe Nature Communications allows space for that. Overall, I find the study reports important results obtained by a comprehensive and innovative methodology. My suggestion is to move some portion of the supplementary information to the main text, so that the results are easier to understand and the central part of the study is more exposed to readers.

We understand your concern regarding the presentation style - It arises mainly from the fact that this manuscript was transferred from its previous submission to 'Nature Geoscience' where the text length

and number of figures were more constrained.

We have expanded the main text of the manuscript from approximately 3160 to 4660 words which now contains more detailed information on and comparison between different methods. At the same time, we expanded the methods section from around 1250 to 2240 words to give the reader the main concepts of the methods while the supplement still provides in a slightly shortened version all the details needed to reproduce our results.

We considered your suggestion of moving more figures from the supplement to the main text but are concerned about interrupting the text fluidity in doing so. We believe that the most important message of the paper is that the combination of various methods provides comprehensive results, rather than the methods themselves which have been applied and described in previous publications. We are also hesitant about moving the figures because we don't want to be emphasising the results from one method more than those from others. Moving the figures would also disrupt the supplementary Information which now (in the revised version) follows a logical order. We are of course willing to do more work in restructuring the manuscript and its supplement but would like to have some guidance from the editor before doing so.

2 Major concerns

1. **Presentation in the main text:** I understand the importance of the study, but I think the implication of the results is not so simple. I am also afraid that the study results do not carry critical and/or immediate impact on the research field. Rather, the careful analysis of the data that obtained by a huge effort in the field provides the basis of the research on the ice sheet dynamics. The main text nicely explains the overview of the study, but excludes necessary and valuable details of methodology and results. I would like to encourage the authors to include text, figures and equations, that are currently presented as supplementary information. For example, Figure 4 in the supplementary information is very important because it explains how vertically mean anisotropy was obtained from the airborne radar data. Moreover, this plot suggests shortcomings of taking the depth-averaged anisotropy. Anisotropy is apparently greater near the bed, which is important and worthwhile for discussion. Similarly, Figure 9 in the supplementary information is essential to describe anisotropy obtained from the phase sensitive radar data. Figure 13 is important as well, because it forms the basis of the enhancement factor calculation. I also like to see Figure 3 (in the supplementary information) in the main text, which shows all the data obtained at the cross sections of the radar profiles (I understand that Figure 2a in the main text shows only a part of the data). It is difficult to say which to be included in the main text, and it is of course a matter of choice by the authors. I wish the authors present more about what you did and obtained, rather than rounding off the details and focusing on relatively uncertain implications.

We have expanded the methods section to provide the information necessary to follow our line of thought without spending extensive time reading the supplement. We also added a paragraph to the results section in which the horizontal anisotropy obtained from different methods is compared and explained where they disagree as well as over which depths we consider our results representative.

We agree that the implications of this work discussed in the paper are not proven with experiments,

but we think the discussion of them raises important questions which need to be addressed in the future. We did qualify some of our statements to emphasise their speculative nature but think that it still adds value in motivating future studies to address these kind of questions.

We are concerned about disrupting the text fluidity of the main manuscript and the supplement when moving more figures from the SI to the main text (see comment above), as well as emphasising the results obtained from one method more than others. The main text focuses on the comparison of the results between methods and their implications for large-scale flow dynamics, which we consider the novelty of this study (all methods have been applied in previous studies which are cited accordingly).

Response to specific suggestions regarding Figures:

- Figure 4 does show the **travel time difference** with depth, **not the depth-dependent anisotropy**. The difference in arrival times naturally increases with depth due to the longer distance travelled, even if the COF remains constant with depth. We used a linear regression through the picked travel times to infer the depth-average anisotropy because of resolution constraints in resolving variations with depth. We made it more clear in the figure caption and the description in the text to avoid a similar misunderstanding by other readers. Thank you for making us aware of it.
- Figure 9 shows the possibility of inferring the depth-varying COF from pRES measurements and indeed is an important figure. But again, the method itself is not new nor is it the focus of this study as it is published and discussed in a separate paper (Zeising et al., 2022).
- Figure 13 shows the modelled correlation between second- and fourth-order spectral coefficients which were used to obtain a full fabric description. It is an important figure for those who are interested in reproducing our results or use the Specfab model to calculate enhancement factors from their own data where only the second-order structure tensor is known. For general readers, however, this is quite a technical figure and would require that extensive explanation would be moved to the main text as well, i.e. the entire section 4 of the supplement.
- Figure 3 indeed shows also the results obtained outside the ice stream which was discarded for simplicity in the figure of the main text because the method essentially was not able to detect anisotropy. For completeness, we decided to add those data points to Figure 2 (in the main text) and added more discussion about the strengths and weaknesses of the individual methods.

As mentioned above, we are willing to make more structural changes if considered necessary by the editor.

2. **Depth-averaged horizontal anisotropy:** I was rather surprised at the procedure to average the anisotropy over the full ice thickness. As clearly seen in Figure 4c in the supplementary information, development of the fabrics is expected in the deeper region. I wonder if it is appropriate to smooth the vertical distribution of the anisotropy. It is actually possible to discuss vertical distribution of the anisotropy using the data (Figure 4c and 9).

We are afraid that there is a small misunderstanding regarding Figure 4c (already mentioned above): it shows the travel-time difference between two polarisation directions oriented more or less ninety degrees to each other. For a fabric with a (hypothetically) constant horizontal anisotropy through the

entire ice column, the difference in the timing of the return signals increases with depth due to the difference in propagation velocity. We are grateful that you pointed out this issue and we adjusted the figure caption in order to avoid a similar misunderstanding by other readers.

Aside from that, the reason to average the anisotropy over depth lies in the large uncertainties that would arise in trying to resolve the depth-variance in anisotropy from the available airborne radar data. The EGRIP ice-core COF shows a relatively constant anisotropy over large parts of the ice column (see e.g. Zeising et al., 2022), based on which we find our averaging approach justified. The pRES measurements allow for reliably resolving depth-related variations in the fabric anisotropy (Figure 9 in the supplement) which is described and discussed in a separate paper (Zeising et al., 2022). We have added a discussion about the differences between methods and the corresponding depth ranges for which they are representative.

3. **Comparison with previous studies** Ice fabrics in ice sheets have been studied by various methods in Antarctica as well as in Greenland. Comparisons with such previous studies are completely missing in the article. How do you compare your results and methodology with those reported in Greenland and Antarctica? It is important to demonstrate how you improve our existing knowledge of ice fabrics and its influence on the ice sheet dynamics.

We have added a section in the discussion where we compare our findings to a previous study conducted at EGRIP, as well as results from radar methods and ice cores from antarctic ice streams. This paper improves our existing knowledge of ice fabrics and their influence on ice sheet dynamics, as it is the first study that, a) covers such a large spatial area, b) compares so many methods and datasets, and c) attempts to translate these horizontal anisotropies directly into flow enhancement factors to link our observations from the field with ice-flow models. This is also emphasised in the introduction.

3 Specific comments

(line numbers in revised version)

- Abstract: The abstract is easy to read for general readers, but I think it is too general as an abstract of a scientific paper. Only two sentences in lines 9-13 are directly related to the finding of the study. The last sentence (application to extraterrestrial ice and other materials) is not very much relevant with the study. I suggest the authors to describe more about the study results, namely ice fabrics and its distributions that you revealed by radar, modelling and the ice cores.

We removed the sentence about extraterrestrial ice bodies in the abstract and the discussion of the implications. The abstract was slightly too long, so unfortunately, we do not see how to add more details about the results while keeping it short enough. Four out of six sentences are now directly related to this paper.

- Line 4: The definition of “fabric” is inconsistent with that given in 19-20.

We changed this to crystal orientation fabric (COF). We also consistently used COF everywhere in the manuscript rather than ‘fabric’.

- Line 88-90: It is very difficult to see significant variations in the anisotropy obtained by beat (Figure

2b). Do you really think it is consistent with the result from the airborne radar?

The beat signature shows the strongest variability in fabric strength in the vicinity of the shear margins. This is also the area where we believe the method to work best, as the orientation of the radar wave polarisation in relation to the fabric axes seems to be optimal there (close to 45 degrees), resulting in strong beat signatures. These beat signatures become weaker when the radar polarisation is aligned with the fabric's principal axis, which we believe is the case in the ice-stream interior. Little horizontal variation in anisotropy strength also leads to the horizontal alignment of the beat signatures with internal stratigraphy, which makes it difficult to distinguish the two, especially in along-flow radar profiles (therefore excluded here).

This polarisation-dependent ability to detect COF anisotropy is also the case for the crosspoint travel-time analysis. However, here the strongest effect is observed when radar waves are polarised parallel to the fabric's principal axes, while the effect is non-detectable when polarised 45 degrees relative to the fabric. In that sense, the two methods complement each other: where we see strong anisotropy with the crosspoint travel-time analysis (e.g. inside the ice stream) the beat signature shows weaker anisotropy or none at all and the other way around (e.g. in the vicinity of the shear margin outside the ice stream).

The observation of increasing anisotropy from outside the ice stream towards the shear margins is arguably difficult to see in Fig 2b because of the scale but is shown in example radargrams in the supplement, Figs. 5-7 (revised Fig.4-6). The increasing anisotropy towards the shear margin is also confirmed by the shear-margin core and the modelling results from Elmer/Ice.

In summary, we believe that the beat-signature results are consistent with other methods within their uncertainties. The weaker anisotropy obtained e.g. inside the ice stream can be explained by the reasons given above. We added more discussion on this topic in the main text (line 98-110) and refer to Figs. 5-7 (revised Fig. 4-6) in the supplement.

- Line 93-95: Which data show this trend? Figure 2a shows no data outside of the shear margins, whereas I see no significant variations in Figure 2b.

This is visible in the example radargrams in the supplement, Figs 5-7 (see also the comment above). We added references to the supplementing Figures at this location in the main text (line 112-116). Fig 2a now also contains the data outside the shear margins. The supplementary information now also contains an additional figure (Fig.9) comparing the different methods across the shear margins. There you can also see that even though the beat signature indicates overall weaker anisotropy near the margins than e.g. Elmer/Ice, it clearly shows a trend towards higher anisotropy nearing the shear margin.

- Line 206-223: Do you think a calving event at the glacier front affects the ice dynamics >500 km inland? I suppose chain reactions, such as “acceleration >> thinning >> steepening >> acceleration >> ...” may propagate inland, but not a single calving event in a decadal or subdecadal time scale. Further, I believe such dynamical response of an ice stream is more affected by horizontal shearing near the side margins as well as vertical shearing near the bed. Thus, numerical experiments are required to investigate the response of the ice dynamics to downstream perturbation. I find the discussion in this paragraph is exaggerated and misleading.

We don't think that processes like calving events at the glacier front of NEGIS affect our survey area directly, which was recently investigated by Grinsted et al., 2022. What we suggest is that the COF alters viscosities and *potentially* upwards propagation in areas closer to the coast. With this discussion, we want to motivate future modelling studies addressing the question of how much further frontal processes propagate with anisotropy and if or not it is important to take into account. We recognise that the original version of the text is misleading and have modified it slightly so that the speculative nature of some statements hopefully becomes more clear (line 342-246) .

- Supplementary information: Texts are very well written. However, some portions should go into the main text, and some others are not necessary. For example, Line 7-23 are Line 30-51 are nice introduction of ice fabrics, but only if this is a dissertation or a text book. I think they are too lengthy and out of the scope of paper supplement. I also suggest to reduce equations, which are appeared in the citing articles.

We removed lines 7–23 and lines 30–51 and shortened the supplement by referring to corresponding literature where possible. We further moved parts to the methods section of the main manuscript and removed some of the equations that can be found in the cited literature and are not crucial to follow our line of thought.

- Line 323-324 in Supplementary Information: Can you clarify the meaning of “close-by shallow core” and “samples are vertical to the ice-core axis”?

Was changed to ”close-by shallow core (≈ 14 km from EGRIP)” and ”All samples were analysed from thin sections that were cut vertically to the ice-core axis ...” (line 398-401)

- Line 472-483 in Supplementary Information: I think this is an important point which should be discussed in the main text.

We moved this part to the methods section in the main manuscript (line 469-482) .

- Figure 5 in Supplementary Information: “Lower panels” are mentioned in the caption, but not appeared in the figure. Mixed up with Figure 6?

Yes, you are right. We've corrected this.

- Line 626-634 in Supplementary Information: Discussion here is rather qualitative. It is not clear what is the conclusion of the calculation.

The purpose of this calculation is to provide context to the obtained enhancement factors from the fabric. It shows that (in addition to its anisotropic nature) the fabric has the potential to alter the ice viscosity by a much larger degree than temperature (the other most important factor besides COF) would under cold-ice conditions. We have clarified this paragraph (line 544-557) .

- Line 639-640 in Supplementary Information: Why “larger viscosity” results in “shorter characteristic time”? It contradicts to Equation (52).

Thank you for pointing out this mistake, you are right of course: Equation 52 is correct and larger viscosities lead to longer characteristic time, i.e. a delayed response. We've adjusted this here and in the main manuscript (line 563-567) .

References

- Grinsted, A., Hvidberg, C. S., Lilien, D. A., Rathmann, N. M., Karlsson, N. B., Gerber, T., Kjær, H. A., Vallelonga, P., and Dahl-Jensen, D. (2022). “Accelerating ice flow at the onset of the Northeast Greenland Ice Stream”. In: *Nature communications* 13.1, pp. 1–4.
- Zeising, O., Gerber, T. A., Eisen, O., Ershadi, M. R., Stoll, N., Weikusat, I., and Humbert, A. (2022). “Inferring horizontal asymmetry of the bulk ice crystal fabric from phase-sensitive radar measurements”. In: *The Cryosphere Discussions*, pp. 1–13.

Author's response to review # 2

January 21, 2023

Dear Dr. Richards,

Thank you for your expert advice and for asking these critical questions. You've raised some important points that we have addressed now and will hopefully be beneficial for the impact of our work. Please find the response to your comments below.

On behalf of the co-authors,
Tamara Gerber

1 Overview

I believe this paper provides insight into fabric development and viscous anisotropy in real-world locations. It uses a novel combination of a wide range of modelling and observational data sets to provide this. In this review, I have focused on the fabric and fabric evolution methods as that is my area of expertise. The text is well written, clear and easy to read throughout. However, in places the manuscript could benefit from being clearer about the assumptions it uses and provide more explanation where results don't agree. I also have some questions about the isotropic fabric modelled downstream of EGRIP. If these are addressed, I believe the manuscript is suitable for publication and will be a very useful contribution to the field.

2 Main comments

(line numbers in revised version)

1. Explanation needs to be given over which depths these fabrics are valid. I would expect them to be valid to a reasonable depth into the ice sheet, but it needs to be explained in the text. How deep does the radar data go? This also links in with the expected fabric downstream as there is unlikely to be significant vertical shear in the top 50% of the ice stream. In general, I think it would be clearer to refer to these results as applying to 'fabrics in the top xx% of the ice stream' throughout the manuscript, as the fabrics very close to the base are likely to be different.

We have added a paragraph in the results section addressing this comment (line: 128-140):

"Additional differences between panels a-c in Fig. 2 might arise due to the depth-averaging nature of our results. The travel-time difference analysis from the airborne data is averaged across the manually picked reflections as deep as possible, though the deepest internal reflections were not visible at each

crosspoint and the bed reflections are often unclear. The results from the automatic analysis of the beat signature are representative of approximately the upper 1700 m of the ice sheet, while reflections in the pRES measurements were only usable in the upper 1500 m of the ice column. Both the travel-time and the beat-signature analyses in theory allow determining the COF variations with depth. However, the vertical resolution of the airborne data is not high enough to resolve vertical COF variations from travel-time differences and the automatic method for deriving the horizontal anisotropy from beat signatures is averaging the horizontal anisotropy across depth using spectrogram analysis. Depth-varying horizontal COF anisotropy could be derived from the travel-time differences in pRES measurements (Supplementary Fig. 8) and from manually picking the beat signature (Supplementary Figs. 4–5). ”

2. For the downstream fabric, I’m confused as to what process causes the fabric to return to isotropic. If a diffusional or regularization process is causing this, is the magnitude of it realistic? [1] finds a pre-factor of approx. $0.3 \sqrt{D:D/2}$ in front of the diffusional term is valid for experimental strain-rates (where D is the strain rate tensor). This value is likely to be lower still at the lower strain-rates at EGRIP. Is this isotropic fabric produced by too much ‘regularization’ in the model?

We understood this trend towards isotropy as being due to down-stream lateral extension, erasing the established vertical girdle with superimposed horizontal single-maximum (i.e. also by lattice rotation). However, on closer inspection, we found a small mistake in the calculation of the strain-rate tensors along the flow line and its correct rotation into the flow-coordinate system. Repeated simulation with the corrected strain-rate (and spin) tensor now show that the downstream fabric now develops a vertical single-maximum (see figure 12 in the revised supplement), which is also consistent with the radar data.

The difference to the previous version is that the vertical girdle disappears more quickly into a vertical single maximum which then again develops into a (weaker) vertical girdle, but the decreasing horizontal anisotropy component is still consistent with radar observations. To help understand the process of this fabric development, we added an additional Figure to the SI (Fig 13) showing the accumulated strain and the eigenvalue development along the downstream flow line. While lateral compression and along-flow extension are the dominant stress regime in the upstream part of NEGIS, the ice doesn’t accelerate more but flows even slightly slower for around 50 km downstream of EGRIP. At the same time, the widening of the ice stream leads to extension perpendicular to the flow direction. Consequently, the c-axes rotate away from the y-direction (horizontal perpendicular to flow) and into vertical, which now is the dominant axis of compression. Around 80 km downstream of EGRIP the flow accelerates again which, similar to the upstream part of NEGIS, leads to along-flow extension and causes the c-axes to move away from the direction of flow. This results again in a vertical girdle although in a weaker form than further upstream. A similar description has been added to the main text (line: 164–172).

You are correct that regularisation in the form of diffusion is used in the model. However, this is spectrally sharpened (hyper diffusion) to conveniently affect only the highest-wave number modes (where small-scale noise in the c-axis distributions is otherwise generated). It can therefore not be directly compared to that used in Richards et al., 2021. The magnitude, however, is also proportional to $\sqrt{D:D/2}$, meaning that (for an appropriately chosen time-step size) the modelled fabric depends only on the total accumulated strain, and not the rate of strain. This is similar but not identical to our Elmer/Ice implementation, in which the rate of diffusion has some temperature dependence. Overall,

we therefore believe that the revised model run is correct and that regularisation does not affect the fabric patterns generated.

3. Supplement line 504: ‘In addition, the spin is assumed to vanish’. This may be the case, but more justification should be provided. If the along flow gradients are small, then even a small amount of curvature could make the vorticity non-negligible. This should be tested, especially as the model can easily accommodate this and the satellite data available contains this information.

This is a good point, and we updated the fabric simulation to include the spin, too, finding no significant change. The revised manuscript includes this change.

4. Line 172 and Fig 3. Why the disagreement between model and radar over easy-to-shear. I don’t think you can say ‘Both radar- and model-based flow enhancement suggest that the ice upstream of EGRIP is easier to shear horizontally than downstream’ as the radar data seems to show no change from isotropic upstream or downstream. There should be some discussion in the text about why the Elmer/Ice results diverge so much from radar data here. This is especially interesting as it appears in Fig 2. That Elmer/Ice and the radar data agree for fabric predictions.

The differences between model and radar-derived enhancements arises from the different COF inferred from the horizontal anisotropy. While the full fabric is known for the modelled data, the radar data only yields horizontal anisotropy, so assumptions about fabric type had to be made to estimate the second-order structure tensor. We have added a few sentences discussing these differences (line: 275–283).

3 Minor comments

- Generally, in the manuscript you use COF or fabric interchangeably, I think it would be clearer to choose one and stick with it.

The revised version uses COF consistently everywhere in the main text and supplement.

- For someone who is not an expert on RES, I would appreciate a bit more explanation of how this process works. For example, I had to look up what birefringent was.

We’ve added a few sentences in the introduction that hopefully prove to be helpful for readers less familiar with RES (line: 44–58).

- Line 59: Here I was a bit confused initially what kind of anisotropy you are referring to? Horizontal anisotropy or viscous anisotropy which comes up later in the manuscript. I think throughout it would be good to always be clear what kind of anisotropy you are referring to.

horizontal anisotropy is right (line: 60). We made sure to clarify this also in the rest of the manuscript/SI.

- Line 129: Similarly, to my main comment 2, you say the stress relaxes, but what deformation causes the c-axes to rotate to vertical symmetry, if there was no deformation then the c-axes would not rotate.

We agree that this is phrased in a slightly misleading way. The stress is not relaxing per se but the deformation regime changes from along-flow extension and lateral compression to lateral extension and predominantly vertical compression. The revised version includes a discussion on how we interpret the

fabric evolution in relation to the accumulated strain (line: 164–172). See also our response to main comment 2.

- Line 161: A sentence or two on these assumptions is needed here, rather than the reader having to refer to the supplement, especially as they are quite strong assumptions. Having looked at the supplement, I agree that they are reasonable, but they should be discussed to some extent.

done (line: 227–235).

- Fig 3. Similarly to above, I don't think it is correct to call a,b,c,d 'measured' enhancement factors, as it relies on some guesses for the eigenvalues. A better subtitle could be 'measurements + approximations' or words to that effect.

~~measured~~ → radar-derived

- Line 175 - should this be a reference to Fig 3.f as well?

We've adjusted this sentence as follows: Both radar-~~and model~~-based flow enhancements suggest that the ice upstream of EGRIP is easier to shear horizontally than downstream (Fig. 3d,e) (line: 261).

- Line 179: flow-direction should not be hyphenated.

adjusted

- Line 193: do you have a reference for the average temperature being -20C?

The thermal model from Elmer/Ice suggests average temperatures of around -20°C, while borehole temperatures from Greenland deep drilling sites show temperatures between -20 and -30°C. Our calculations are relatively insensitive to the exact choice of T_1 as long as the assumption of ice being at cold conditions (i.e. below -10°C) holds. We therefore changed the 'assumption that the surrounding ice is at -20 °C' to 'assuming cold-ice conditions' (line: 290) but added corresponding references in the supplement (line: 539-542).

- Line 300 and in the supplement: There should be more discussion on the equivalence/difference between the fabric evolution models, and on the Specfab model, As I had to read [2], and the github page to understand. Does Specfab include a diffusional term?

We have updated the text (line: 483–489 main text) and (line: 397–406 in the supplement) to note that our fabric evolution model is identical to that used in Elmer/ice, apart from the regularisation (diffusion). The model is simply posed in spectral space instead of tensorial space as in Elmer/Ice. The regularisation (diffusion) is concentrated in the highest modes in spectral space (see above point), which is truncated at sufficiently high wave modes to avoid its effect on the lowest-order modes (i.e. to avoid it affecting $a^{(2)}$ and $a^{(4)}$ orientation tensors). The below figure shows how Specfab compares to Elmer/Ice in 3D ice-cube-crushing experiments using the current anisotropy module in Elmer/Ice (lattice rotation module). We find the two models compare excellently (and to some data; not shown), but will refrain from showing this figure in the paper, as it will be published soon in another paper by D. Lilien where he also considers recrystallisation processes. We hope you agree that it suffices to discuss these experiments but not show them in detail (see Section 2.2 of the revised SI).

Figure 1: $a^{(2)}$ eigenvalues for four modes of deformation. Blue, orange and green indicate the largest, mid, and smallest eigenvalues of $a^{(2)}$. Simulations were run until 100% strain, except for the simple-shear experiment that was run until 300% shear strain.

References

Richards, D. H., S. S. Pegler, S. Piazzolo, and O. G. Harlen (2021). “The evolution of ice fabrics: A continuum modelling approach validated against laboratory experiments”. In: *Earth and Planetary Science Letters* 556, p. 116718.

Author's response to review # 3

January 23, 2023

Dear reviewer,

Thank you for taking the time to read our work and for your detailed comments, which pointed us towards potentially unclear parts of the manuscript. We hope the implementation of your feedback in the revised version increases the clarity and lead to an overall improvement of this work. Please find the point-to-point response below.

On behalf of all co-authors,

Tamara Gerber

1 Main comments

This paper investigates how the ice fabric translates to mechanical anisotropy of ice in the onset region of the NEGIS, Greenland. The study uses a set of independent techniques comprising of radar sounding, ice core analysis and numerical modelling. The noteworthy results are the maps of measured/modelled flow enhancement factors for pure shear compression/extension and horizontal shear deformation in the NEGIS region. These are depicted in Fig. 3 of the main manuscript. Under assumptions made by the authors, their results show that the ice in the upstream portion of NEGIS is an order of magnitude harder than isotropic ice further downstream. Also, the upstream ice is around three times harder than ice outside the ice stream. Data is also presented on depth-averaged fabric distribution and flow enhancement factors. The work is significant to the radioglaciology community and it builds upon established literature that have used some of the present techniques in other ice stream areas. The relevant references for such have been appropriately cited. To the best of my knowledge, the work presented in this paper is the first for this dynamic region of Greenland.

The claim in the abstract and conclusion on transferring the methodology to other planets and geologic materials requires additional supporting evidence. It is currently not a well-justified claim not least because the radar and coring technologies for extra-terrestrial ice characterisation will differ considerably to those applied in Greenland in this paper. At the very least, some discussion on how the authors expect their technique can transfer to extra-terrestrial ice characterisation and what the limitations are could strengthen their claim. Clarification should also be made for the ice inside and outside the ice stream: in the abstract it is claimed to be two times harder, but the main text suggests three times harder inside.

We removed the statements of potential relevance for extraterrestrial ice bodies from the abstract/conclusion. Elaborating more on transferring these applications to other planets/satellites would be out of scope of this work. The softening by factor two in the abstract refers to the susceptibility of the *shear margins* towards horizontal *simple shear* deformation while factor 3 in the main text refers to *pure shear* (i.e. extension/compression) *inside the ice stream*. We tried to clarify this in the main text.

The data analysis and interpretation on the whole are sound. I would have like to see more detail in the comparison with the different methods used i.e. the radar sounding, ice flow modelling and ice core analysis. The presentation of the data analysis in the form of 2D maps does not make clear how the different techniques support each other and to what extent.

We expanded the results section with more comparison between the results obtained with the different radar methods, including the strengths and weaknesses of each, as well as over which depths we consider the corresponding results representative (line 120–140 in the revised manuscript) .

I consider the authors' methodology is sound. Some of plots such as the radargram are missing key scalebars which are normally expected to be included. Another minor detail I found to be missing is the explicit GPS coordinates of the cross-point locations for part of the work to be reproduced.

The scale bars were added to the radargrams and GPS points were included in table 2 in the SI. The data files including the crosspoint analysis will be made available which also includes GPS information for all the data points.

In summary, I enjoyed reading your paper and its supplementary. Please find attached some minor corrections/suggestions that may help improve your manuscript. In general, I found it well written but a little disjointed in places especially in the supplementary, which currently reads as if different sections have been written by different authors but without an overarching edit that joins them up consistently. As you will see in my comments, there are many areas where I felt specific quantification are lacking in the expressions used to describe your results and observations. These could be tightened up.

We shortened and edited the supplement so it hopefully reads with more fluidity and clarity. Our response to your comments in the attached files can be found below.

It is impressive that you have been able to use multiple independent techniques to arrive at the key results for this region of Greenland. While the maps in Fig. 3 are useful and key, I believe that an additional simple 2D plot directly comparing the results from the different radars, ice flow modelling and ice core analyses would be more meaningful. For instance, an X-Y plot of the $\Delta\lambda$ versus flow direction along the centre of the ice stream. Such a plot could overlay the results derived from the radars, modelling and ice cores – which would greatly strengthen the support of each other. It is hard, as a potential reader, to see/compare in any quantitative detail the differences in results from the different techniques from the existing 2D maps produced.

We have added a figure to the supplement similar to what you're suggesting. However, instead of showing the $\Delta\lambda$ values along the centre of the ice stream, we find a 2D plot across the ice stream more meaningful (Fig. 9 in the revised SI). However, one has to bear in mind the differences in depth ranges over which the corresponding methods are representative, as well as the differences between absolute

and apparent (lower-bound) horizontal anisotropy, which is discussed in the accompanying text to the figure (line 260–289 in the revised supplement) .

I would also recommend some additional discussion on the asymmetry, if any, of the fabric observed on both shear margins. From the Fig. 1, the extinction node lines are visible in the radargram (highlighted in box d) but on the opposite shear margin it is less pronounced.

Unfortunately, our methods do not allow us to directly 'observe' the fabric in the shear margins due to loss of radar return power. The closest observation we have is that the beat signatures in radar profiles perpendicular to the ice-flow direction indicate increasing horizontal anisotropy towards the shear margins, in particular outside the ice stream. The region outside the southeastern shear margin in profile c'–c" in Fig.1 is characterised by heavy folding of deep ice units. Similar to the folds in the shear margins, these features cause a partial loss of the return signal due to steep layer inclination. We interpret this as the main reason for the seemingly weaker beat signature outside the shear margin in that area. Further downstream, beat signatures of similar strength are observed around both shear margins. We have added a sentence on this in the results section (line 102–107 in revised manuscript) .

2 Response to attached file 'ref #3 MS'

(Line numbers referring to the original manuscript)

- line 7-8: ~~methods applied to extensive~~
- line 15: we removed this sentence
- line 15-17: We removed this sentence due to the constraints in abstract length and because the relevance of extraterrestrial ice bodies is not the main focus of this paper (see also comment by reviewer #1)
- line 27: adjusted to be plural in the previous sentence
- line 28: ~~thorough~~ → physical
- line 31: ~~not isotropic~~ → anisotropic
- line 32: errors *of yet poorly understood magnitude*
- line 36: ~~ice streams~~ → ice-stream dynamics
- line 37: fast flow ($> 10 \text{ ma}^{-1}$)
- line 41: ~~years ago~~ → more than a decade ago.
- line 42: However, both the application of these models to ice streams and *spatially extensive* in-situ observations of the COF *in these dynamic regions remain rare*.
- Fig. 1: flow lines were removed, spelling corrected
- line 43: ~~by being~~ → to
- line 44: ~~uncertain~~ → unknown

- line 46: ~~unlikely to represent~~ → unrepresentative of
- line 55: ~~increased/decreased~~ → maximum/minimum
- line 56: ~~anomalies~~ → differences, Δt ; ~~different~~ → orthogonal
- line 59: distribution of *the horizontal* anisotropy
- line 60/61: using an extensive data set of radar measurements with various radar systems and a combination of independent methods.
- line 65: deformation mechanism and its direction relative to the COF principal axes.
- line 74: There was no ground-based profile downstream of EastGRIP available at the time of analysis.
- line 78: ~~understanding~~ → interpretation
- line 87: This is shown in Figs. 5-7 in the supplementary information.
- line 89: weaker ~~results~~ horizontal anisotropy, which ~~are potentially~~ may be
- line 90: ~~as well as~~ → and
- line 92: width of 5–10 km *and throughout the entire ice column but the top $\approx 10\%$*
- line 93: ~~that region~~ → these regions.
- line 107: We consider a size large enough to statistically represent the COF → ice parcel (*large enough to statistically represent the COF*).
- line 117: is *much* smaller
- line 119: Due to the lack of signal return power across large depths of the shear margins,
- line 122: Outside the NEGIS, the pRES measurement indicates a ~~decreasing~~ *decreased* anisotropy *compared to inside the ice stream*, which agrees with the results from the beat-signature analyses and the Elmer/Ice model *at the corresponding location*.
- line 127: Downstream of EGRIP the ice stream widens, ~~and the applied stress relaxes due to a decrease in flow acceleration and compression perpendicular to ice flow~~ *leading to lateral extension while the along-flow extension decreases due to a relatively constant flow velocity which accelerates again around 80 km downstream of EGRIP*.
- line 155: likely *to be true*
- line 163: ~~with respect to~~ → for
- line 167: ~~frequency~~ *signature*
- line 169: Beat-signature and travel-time analysis show a trend towards slightly softer ice downstream of EGRIP ($E_{xx} \approx .02$) compared to further upstream ($E_{xx} \approx .1$).
- Figure 3: updated figure in the revised version.
- line 186: from ~~current methods~~ the available nadir-pointed radar data.

- Figure 4, caption: in the stupper part of the NEGIS upstream half of the survey region inside the NEGIS
- line 199: over *short* horizontal distances of a few kilometers
- line 219: full-Stokes
- line 229: ~~can not~~ → cannot
- line 231-237: we removed this part
- line 249: *each* functioning as *a* transmitter and *a* receiver
- line 254: the received ~~signal was~~ *signals were* sampled at a frequency of 1.6 GHz. The airplane ~~was flying~~ *flew* at an approximate altitude...
- line 261: ~~distance~~ → area
- line 268: ~~antenna in horizontal direction~~ antennas
- line 269: *each* ranging from
- line 274: ~~in~~ → into, ~~rays~~ → components
- line 277: ~~the~~
- line 285: ~~main principle~~ → principal
- line 287: τ
- line 288: *point* location
- line 293: the model accounts for lattice rotation in 3D, does not take into account recrystallization processes.
- line 294: dimensions specified in the revised version.
- line 303: defined above.
- line 311: ~~tensor~~ → factors
- line 322: ~~the~~ → another

3 Response to attached file 'ref #3 SI'

(Line numbers refer to the original manuscript)

- line 7: we removed this part
- line 77: *At typical radar frequencies in the range of 1 MHz to 1 GHz, ...*
- Fig.1, caption: ~~crosspoins~~ → crosspoints
- line 103: ~~signal of radar~~

- line 105: ~~in~~ → into
- line 106: ~~rays~~ → wave components
- line 112: ~~different~~ → near-orthogonal
- line 121: *second-order* spline interpolation
- line 128: However, the instrument sensitivity of the DEP device is not high enough to measure these effects *and permittivity variations below the transition into pure ice is related to instrument noise* (Wilhelms et al., 1998). To estimate the radar wave speeds ~~while eliminating instrument noise~~, we assume a firm permittivity of 1.55 (Mojtabavi et al., 2022) at the ice-sheet surface and extrapolate the smoothed DEP profile (moving average with 5 m window length) from the core onset (13 m depth) to the surface. Below the transition into pure ice at ~ 200 m we assume a constant permittivity of 3.15.
- line 139: affect the ~~analysis of the~~ *calculation of reflector depths at CPs* in the immediate vicinity of the shear margins.
- Figure 2: corrected
- line 140: *the spatially uniform* wave speed assumption
- line 148: Crosspoints τ where less than five reflections could be clearly identified τ are excluded from further analyses.
- Figure 3: we removed this figure since the whole dataset is now shown in Fig.2 in the main text.
- Table 1, caption: ~~travel-time anisotropy~~ → difference
- Table 1: units were added
- line 185: ~~compared~~
- Table 2: GPS coordinates were added ; ~~crosspoints~~ → crosspoints
- line 187: The maximum distance between the parallel-flow and the across-flow radar traces is 11.7 m ~~at maximum~~
- line 205: waves at ~~slightly different frequencies or wavelengths~~
- line 227: ~~out of phase~~ → anti-phase
- line 228: ~~uneven~~ → odd
- line 232: normally λ_{mod} would be used for wavelength. Yes, but λ is traditionally also used for eigenvalues. To avoid confusion with eigenvalues derived from the beat signature we decided to keep l_{mod} as symbol for beat wavelength.
- line 239: is
- line 243: the upper ≈ 1500 m
- line 246: Manual → Semi-manual

- line 247: ~~fabrie asymmetry~~ → COF anisotropy
- line 256: ~~can also allows to~~ determine
- line 260: manual → semi-manual
- line 263: ~~an automated~~ → a semi-automated
- line 270: new paragraph, ~~then~~ → first
- line 277: corrected for artefacts by correcting or removing wrongly determined dominating modes (see red line in the bottom panel of Fig.S6)
- line 279: ~~is very efficient~~ → allows to process and analyse the radargrams automatically, thus reducing the required time.
- Figure 5: gray scale added
- Figure 5, caption: The signature shows an artefact in the upper 700 m from the source chirp; ~~Lower panels~~ (Gardner et al., 2021)
- line 281: the limitations are in parts ambiguities in the spectral analysis *to determine the dominating mode*, for instance, if internal layers are spaced at a vertical distance comparable to the beat signature.
- Figure 6: gray scales have been added
- Figure 6, caption: both shear margins (*visible around km 13 and 36*); the part of this profile northwest of EGRIP
- line 311: ~~anisotropy~~ → difference
- Figure 7: scale bars were added.
- Figure 8, caption: ~~as see, Figs. 7, 6 and 5~~ → Figs. 5, 6 and 7
- Figure 9, caption: ~~The~~ → These, ~~showed here~~
- line 482: ~~well enough~~ → within the methods uncertainties. Note that this paragraph was moved to the main file.
- line 493: ice parcel (*large enough to statistically represent the COF*)
- line 504: The spin tensor, \mathbf{W} , is calculated from the surface velocities, assuming the vertical components are zero:

$$\mathbf{W} = \frac{1}{2} \left(\tilde{\nabla} \mathbf{u} + (\tilde{\nabla} \mathbf{u})^T \right) = \mathbf{0}. \quad (1)$$
- line 601: ~~full~~ → brown and petrol solid lines
- line 605-606: fabric samples from the EGRIP ice core (~~red squares~~) → (purple triangles) agree well with the modelled correlation, as does the NEGIS shear margin ice core ~~red triangles~~ → (purple squares)
- line 607: ~~full~~ → brown and petrol solid line
- line 608-609: ~~dotted light grey~~ → red solid line, ~~dashed dark grey line~~ → black solid line

- Figure 14: (both from the crosspoint analysis (dots with white rim) and beat frequency anisotropy analysis (without rim)). Similar adjustments in Figs. 15, 16, 18, 19 and 20.
- line 640: ~~Erge~~ → Therefore

References

- Mojtabavi, S., O. Eisen, S. Franke, D. Jansen, D. Steinhage, J. Paden, D. Dahl-Jensen, I. Weikusat, J. Eichler, and F. Wilhelms (2022). “Origin of englacial stratigraphy at three deep ice core sites of the Greenland Ice Sheet by synthetic radar modelling”. In: *Journal of Glaciology*, pp. 1–13. DOI: [10.1017/jog.2021.137](https://doi.org/10.1017/jog.2021.137).
- Wilhelms, F., J. Kipfstuhl, H. Miller, K. Heinloth, and J. Firestone (1998). “Precise dielectric profiling of ice cores: a new device with improved guarding and its theory”. In: *Journal of Glaciology* 44.146, pp. 171–174. DOI: [10.3189/S002214300000246X](https://doi.org/10.3189/S002214300000246X).

REVIEWERS' COMMENTS

Reviewer #1 (Remarks to the Author):

Review comments on the revised version of "Crystal fabric anisotropy causes directional hardening of the Northeast Greenland Ice Stream" by Gerber et al.

Suggestions:

The paper is improved by revision. I acknowledge the effort of the authors to respond to the comments and suggestions given from different aspects. I still have an issue with the presentation style, i.e. main text with four figures is supported by Supplementary Information (SI) with 21 figures. However, the authors have made effort for this point, by including more descriptions in the main text. Judging from the fact that no other reviewer has a problem with it, I should stop complaining.

Reading this paper is not easy for me because I have to go forth and back between the main text and SI. It is OK if paper presents simple properties (e.g. temperature, ice thickness, melt rate ...). I can read the main findings first, and then learn later how these properties were measured or derived. However, because Figures 2 and 3 are not that kind, it was necessary for me to go to SI before reading the main text and catching the message of the paper.

To help readers like me, I suggest the authors referring the section and figures of SI more frequently in the main text. For example, travel-time difference and beat analyses are mentioned in line 54-58. It is helpful if readers are informed that details of these techniques are given in SI Sections 1.1 and 1.2. The same is true for the Elmer/Ice and Specfab models mentioned in lines 85-86, which are described in SI Section 2. Another example is the enhancement factor introduced in line 204 and explained in SI Section 3.

SI is referred to more in the Method section. However, you can improve by clearly connecting "Travel-time difference and beat-signature analyses" to SI Section 1, and "Modelling of COF evolution" to SI Section 2, for example.

I also suggest referring to figures more frequently in the text. For example, I assume most of the text on page 5 explains data presented in Figure 2. Nevertheless, it is not clear which panel of Figure 2 is mentioned in each sentence. Indicating exact figure numbers, e.g. (Fig. 2a), (Fig. 2c), is helpful to follow the description of the data. The same is true for lines 236-243 and line 262-, where not clear which panel of Figure 3 is mentioned.

Minor comments:

line 235: Here and in other places, I find "Supplementary Method". Should it be "Supplementary Information"?

lines 251 and 338: isotropy >> isotropic ice?

line 482: Section 3.1 >> Section 2.1?

Review comments on the revised version of "Crystal fabric anisotropy causes directional hardening of the Northeast Greenland Ice Stream" by Gerber et al. (specific to the revision addressed to comments from Referee #2)

The comments from Referee #2 were focused on ice fabrics and its development. The authors responded the technical comments, mostly by adding text to provide more information. I find the referee's concerns are adequately addressed and the manuscript is benefited by the revision. I have only one comment and one minor suggestion.

To address the referee's second comment ("what process causes the fabric to return to isotropic?"), the authors discuss the flow regime in the vicinity and downstream of EGRIP (lines 164-172). The authors describe "as the COF transitions from a vertical girdle to a vertical single maximum roughly 50 km downstream of EGRIP as a consequence of ice-stream widening and the associated combination of flow-transverse extension and decreased along-flow acceleration" (line 165-168). It is true that fast-flowing area is widening from EGRIP to its lower reaches. However, widening of fast-flowing area does not mean that ice is diverging to lateral direction (transverse extension). Do you see such a divergent flow regime in the surface flow vectors? Actually, if I understand it correctly, the "flow tubes" in the Elmer model show that ice flow laterally converges in the vicinity of EGRIP, while fast flowing region is widening downglacier (Fig. 3c).

Minor comment:

line 164-165: "The modelled COF evolution of this simple experiment" >> The COF evolution obtained by this simple experiment?

Reviewer #3 (Remarks to the Author):

Many thanks for addressing my comments/suggestions in detail. The revised manuscript has been improved and the clarity is also much better from a reader's point of view.

Below is a list of (very) minor corrections/clarifications that could be considered:

- * Line 106: Further downstream : how much further exactly do you mean?
- * Line 174: vertical girdle ($\Delta_{\lambda} = 0.6$). Instead, do you mean "...horizontal anisotropy ($\Delta_{\lambda} = 0.6$) agrees with a vertical girdle"?
- * Line 186: [5, 14]
- * Line 294: [7, 31]
- * Line 215: what does "coaxial" mean? perhaps reword it.
- * Line 216: Do you really mean to say "eigenenhancements" ?
- * Fig. 3 last sentence of caption is duplicated in Line 220-222 of main text. Consider removing one or the other.
- * Line 276: some previous -> prior
- * Line 231: I do not understand what "one horizontal" means.
- * Line 254: please quantify your definition of far-field of the ice stream
- * Lines 307-9: [5, 46] and [14, 16, 17]
- * Line 329: based
- * Line 395: Stewart
- * Line 399: what is meant by "follow the same principles"? sample preparation and measurement are different to my understanding.
- * Line 405: COF observations -> COF measurements
- * Line 412: different wave polarisations -> different polarised waves
- * Line 419: 450 m to 3000 m

* Line 427, 437, 438: angle unit definitions have now changed to radians. Stick to degrees for consistency throughout manuscript.

* Line 447: do you mean "thus does not" ?

* Line 503: [7, 31]

Point-by-point response to reviewer comments

March 25, 2023

1 Reviewer # 1

The paper is improved by revision. I acknowledge the effort of the authors to respond to the comments and suggestions given from different aspects. I still have an issue with the presentation style, i.e. main text with four figures is supported by Supplementary Information (SI) with 21 figures. However, the authors have made effort for this point, by including more descriptions in the main text. Judging from the fact that no other reviewer has a problem with it, I should stop complaining.

Reading this paper is not easy for me because I have to go forth and back between the main text and SI. It is OK if paper presents simple properties (e.g. temperature, ice thickness, melt rate ...). I can read the main findings first, and then learn later how these properties were measured or derived. However, because Figures 2 and 3 are not that kind, it was necessary for me to go to SI before reading the main text and catching the message of the paper.

To help readers like me, I suggest the authors referring the section and figures of SI more frequently in the main text. For example, travel-time difference and beat analyses are mentioned in line 54-58. It is helpful if readers are informed that details of these techniques are given in SI Sections 1.1 and 1.2. The same is true for the Elmer/Ice and Specfab models mentioned in lines 85-86, which are described in SI Section 2. Another example is the enhancement factor introduced in line 204 and explained in SI Section 3.

We have added the corresponding references in the suggested locations.

SI is referred to more in the Method section. However, you can improve by clearly connecting "Travel-time difference and beat-signature analyses" to SI Section 1, and "Modelling of COF evolution" to SI Section 2, for example.

We have added a sentence in the first paragraph of the methods section to give an overview: Details on each method, validation and uncertainties are presented and further discussed in the Supplementary Information: The analytical methods of travel-time and beat-signature analyses are described in Section 1, model details and the performed simulation to obtain the modelling results are described in Section 2, the calculation of enhancement factors and the underlying assumptions are detailed in Section 3, temperature difference estimates corresponding to enhancements caused by COF are further described in Section 4, and an estimate of the characteristic time is shown in Section 5.

Further down in the Methods section there are additional, more specific references to the Supplementary Information.

I also suggest referring to figures more frequently in the text. For example, I assume most of the text on page 5 explains data presented in Figure 2. Nevertheless, it is not clear which panel of Figure 2 is mentioned in each sentence. Indicating exact figure numbers, e.g. (Fig. 2a), (Fig. 2c), is helpful to follow the description of the data. The same is true for lines 236-243 and line 262-, where not clear which panel of Figure 3 is mentioned.

We added figure references in lines 79, 80, 81, 87, 96, 168, 193, 237, 241, 254, 259, 266, 268, 283.

line 235: Here and in other places, I find "Supplementary Method". Should it be "Supplementary Information"?

I think you're right, we changed it to 'Supplementary Information' here and also in lines 286, 368, 482, 497, 514.

lines 251 and 338: isotropy → isotropic ice? we followed your suggestion in both places.

line 482: Section 3.1 → Section 2.1? Yes, you are right, we've changed this.

2 Reviewer # 2

The comments from Referee #2 were focused on ice fabrics and its development. The authors responded the technical comments, mostly by adding text to provide more information. I find the referee's concerns are adequately addressed and the manuscript is benefited by the revision. I have only one comment and one minor suggestion.

To address the referee's second comment ("what process causes the fabric to return to isotropic?"), the authors discuss the flow regime in the vicinity and downstream of EGRIP (lines 164-172). The authors describe "as the COF transitions from a vertical girdle to a vertical single maximum roughly 50 km downstream of EGRIP as a consequence of ice-stream widening and the associated combination of flow-transverse extension and decreased along-flow acceleration" (line 165-168). It is true that fast-flowing area is widening from EGRIP to its lower reaches. However, widening of fast-flowing area does not mean that ice is diverging to lateral direction (transverse extension). Do you see such a divergent flow regime in the surface flow vectors? Actually, if I understand it correctly, the "flow tubes" in the Elmer model show that ice flow laterally converges in the vicinity of EGRIP, while fast flowing region is widening downglacier (Fig. 3c).

The fabric evolution along a flowline downstream of EGRIP was calculated with the specfab model, where the COF is iteratively updated from the spin and strain rate tensors in the corresponding flow-line segment. The modelled flow line is relatively straight, the spin is consequently small. Therefore, the COF evolution is effectively a result of the cumulative strain, while other processes affecting the crystal orientation are not taken into account here. Both the strain rate and spin tensors were calculated from satellite-based surface velocities (see Equation 34–36 in the Supplementary Information), and the cumulative strain along the flow line is shown in the Supplementary Figure 13. From this figure it becomes apparent that the cumulative strain in y-direction (transverse to flow) increases along the first part of the flowline downstream of EGRIP (positive strain = extension).

To answer your question: yes we do observe this transverse flow in the surface flow field, in particular around 40 km downstream of EGRIP. A possible reason for this could be changes in the basal slipperiness and topography, e.g at the onset of an overdeepened trough ~ 40 km downstream of EGRIP, which can cause small deviations in the flow pattern. The flow tubes used in Elmer/Ice only extend to 40 km downstream and due to its proximity to the model boundaries, results obtained from Elmer/Ice in this area are less trustworthy than further upstream. Nevertheless, a small decrease in the horizontal eigenvalue difference in the ice-stream center between EGRIP (0.66) and the model boundary (0.54) is also obtained with Elmer/Ice where flow tubes diverge slightly (flow-line distance at EGRIP = 2.513 km, flow-line distance at model boundary = 3.015 km).

You raised an important point that the ice-stream widening is not necessarily coupled with, or the reason for divergent flow, although both is observed in the first 40 km downstream of EGRIP. We modified the above-quoted sentence as: "The COF evolution obtained by this simple simulation shows a gradual decrease of the horizontal anisotropy, as the COF transitions from a vertical girdle to a vertical single maximum roughly 50 km downstream of EGRIP as a consequence of ~~ice-stream widening and the associated combination of~~ flow-transverse extension and decreased along-flow acceleration (Fig. 2e). "

Minor comment: line 164-165: "The modelled COF evolution of this simple experiment" \rightarrow The COF evolution obtained by this simple experiment?

The modelled COF evolution of this simple experiment" \rightarrow The COF evolution obtained by this simple simulation

3 Reviewer # 3

Many thanks for addressing my comments/suggestions in detail. The revised manuscript has been improved and the clarity is also much better from a reader's point of view.

Below is a list of (very) minor corrections/clarifications that could be considered:

- Line 106: Further downstream : how much further exactly do you mean? We've replaced this sentence as follows: ~~Further downstream, beat signatures of similar strength are observed around both shear margins.~~ \rightarrow Radar profiles crossing the ice stream 20 km further downstream and beyond show beat signatures of similar strength around both shear margins
- Line 174: vertical girdle ($\Delta\lambda = 0.6$). Instead, do you mean "...horizontal anisotropy ($\Delta\lambda = 0.6$) agrees with a vertical girdle"? Clarified as follows: ~~Our Elmer/Ice modelled and radar derived horizontal anisotropy near the EGRIP drill site, in particular with the travel-time method applied on pRES and airborne crosspoint data ($\Delta\lambda = 0.4$), agrees with a vertical girdle observed over large depths in the EGRIP ice core ($\Delta\lambda = 0.6$) [32].~~ \rightarrow Near the EGRIP drill site, the Elmer/Ice-modelled ($\Delta\lambda = 0.66$) and the radar-derived horizontal anisotropy, in particular with the travel-time method applied on pRES and airborne crosspoint data ($\Delta\lambda = 0.4-0.55$), agrees well with a vertical girdle observed over large depths in the EGRIP ice core ($\Delta\lambda = 0.6$) [32].
- Line 186: [5, 14] done.
- Line 294: [7, 31] done.

- Line 215: what does "coaxial" mean? perhaps reword it. coaxial in the sense of parallel to the COF coordinate system. We rephrased the sentence as follows: The enhancement factors presented here are therefore calculated assuming the strain-rate tensor is oriented in alignment with the COF principal frame.
- Line 216: Do you really mean to say "eigenenhancements" ? Yes, this is to emphasize that they are calculated in the COF 'eigenframe'.
- Fig. 3 last sentence of caption is duplicated in Line 220-222 of main text. Consider removing one or the other. Thanks for pointing this out. We've shortened the last sentence in the figure caption so it reads as: The term 'along-flow' therefore refers to the direction of the smallest horizontal eigenvalue.
- Line 276: some previous → prior adjusted as suggested.
- Line 231: I do not understand what "one horizontal" means. this refers to one horizontal eigenvalue. We clarified this sentence as follows: ~~The horizontal single maximum observed in the S5 core and the Elmer/Ice modelling results lead to the assumption that one horizontal and the vertical eigenvalue are of similar size in the shear zone.~~ → The horizontal single-maximum observed in the S5 core and the Elmer/Ice modelling results lead to the assumption that one horizontal eigenvalue is of similar size as the vertical eigenvalue in the shear zone.
- Line 254: please quantify your definition of far-field of the ice stream We agree that this is a rather vague description of the modeled flowtubes far away from the shear margins. We rephrased the sentence as: ~~Enhancement factors from Elmer/Ice increase from 0.3 in the far-field of the ice stream towards 1.0 just outside the ice stream margins, indicating an increasing susceptibility to shear deformation.~~ → Enhancement factors from Elmer/Ice outside the ice stream generally are smaller in the outer flow tubes (0.3–0.5) than in the central ones (0.8–1.4) and increase within the individual model domains towards the shear margins.
- Lines 307-9: [5, 46] and [14, 16, 17] done.
- Line 329: based ~~basedd~~ → based
- Line 395: Stewart ~~Steward~~ → Stewart
- Line 399: what is meant by "follow the same principles"? sample preparation and measurement are different to my understanding. Here, we're referring to the processing procedures that are identical for both ice cores (EGRIP and S5). Hopefully, this is clarified now: ~~The preparation and measurement of the samples follow the same principles.~~ → The preparation and measurement of the samples from both ice cores follow the same procedures: All samples were analysed from thin sections that were cut vertically to the ice-core axis and have dimensions of about 90×70×0.3 mm³.
- Line 405: COF observations → COF measurements ~~Here, we use the COF observations to validate our observations and modelling results.~~ → Here, we use the COF measurements to validate our geophysical observations and modelling results.
- Line 412: different wave polarisations → different polarised waves we'd prefer: Varying wave speeds for 'waves polarised in different directions'

- Line 419: 450 m to 3000 m done.
- Line 427, 437, 438: angle unit definitions have now changed to radians. Stick to degrees for consistency throughout manuscript. Thanks for pointing this out, we've adopted your suggestion.
- Line 447: do you mean "thus does not" ? We've slightly modified this sentence for simplification: *Consequently, the inability of these methods to detect horizontal anisotropy does not prove the absence thereof.*
- Line 503: [7, 31] done.